



# Synchronized Helix Wake Mixing Control

Aemilius A. W. van Vondelen[1], Marion Coquelet[1], Sachin T. Navalkar[2], and Jan-Willem van Wingerden[1]

[1]Delft Center for Systems and Control, Delft University of Technology, Delft, the Netherlands
[2]Siemens Gamesa Renewable Energy B.V., the Hague, the Netherlands

**Correspondence:** Aemilius A. W. van Vondelen (a.a.w.vanvondelen@tudelft.nl)

**Abstract.** Wind farm control optimizes wind turbines collectively, implying that some turbines operate suboptimally to benefit others, resulting in a farm-level performance increase. This study presents a novel control strategy to optimize wind farm performance by synchronizing the wake dynamics of multiple turbines using an Extended Kalman Filter (EKF)-based phase estimator in a Helix control framework. The proposed method influences downstream turbine wake dynamics by accurately estimating the phase shift of the upstream periodic Helix wake and applying it to its downstream control actions with additional phase offsets. The estimator integrates a dynamic Blade Element Momentum model to improve wind speed estimation accuracy under dynamic conditions. The results, validated through turbulent large-eddy simulations in a three-turbine array, demonstrate that the EKF-based estimator reliably tracks the phase of the incoming Helix wake, with slight offsets attributed to model discrepancies. When integrated with the closed-loop synchronization controller, significant power enhancement with respect to the single-turbine Helix can be attained (up to +10% on the third turbine), depending on the chosen phase offset. Flow analysis reveals that the optimal phase offset sustains the natural Helix oscillation throughout the array, whereas the worst phase offset creates destructive interference with the incoming wake, which appears to negatively impact wake recovery.

## 1 Introduction

Optimizing wind farm layout is an important aspect of wind farm design (Manwell et al., 2010). Specifically, the spacing between wind turbines, typically ranging from 3 to 7 rotor diameters, has a strong influence on power production and turbine fatigue life due to the presence of wake interactions between turbines. When the wind direction aligns with the turbine array, downstream turbines can experience a performance drop of up to 20% in some wind farms (Barthelmie et al., 2009). However, increasing the spacing between turbines requires more offshore sea and cabling, reducing the energy density and increasing the cost of the power production site. Therefore, a site-specific trade-off is often made, balancing cost optimization with accepted losses due to wake effects.

To mitigate these losses in existing and future wind farms, researchers are exploring innovative control solutions to recover the 'lost' energy. One prominent method is wake steering, which involves intentionally misaligning the upstream turbine's rotor to redirect the wake away from downstream turbines (Fleming et al., 2014). While promising, this technique only slightly reduces the wake intensity and may still impact other turbines further downstream in the farm. Also, if the wake is not deflected enough, it can still partially impinge on downstream turbines, which can cause load increases. Recent advancements in optimiz-





ing farm layouts using control co-design with wake steering have shown potential for further enhancing production (Baricchio et al., 2024; Stanley et al., 2023).

Another category of wind farm control approaches focuses on wake mixing, which aims to enhance the mixing of the wake with the surrounding free stream flow. The earliest method, dynamic induction control (DIC), involves dynamically varying the
turbine thrust, either by adjusting torque or pitch, to create a pulsating wake that mixes more rapidly with the free stream (Goit and Meyers, 2015; Frederik et al., 2020b). This approach has demonstrated significant power gains, albeit with increased tower load variations and harmonics in grid power. A later method involves rotating the thrust vector across the rotor disk, creating a helical wake shape (Frederik et al., 2020a). Compared to DIC, the Helix approach results in lower tower load variations and higher power gains, garnering considerable attention (Frederik and van Wingerden, 2022; van Vondelen et al., 2023a).

Most studies have applied the Helix approach to an upstream turbine, maintaining baseline control for the downstream turbines. However, in multi-turbine arrays, applying the Helix approach to multiple turbines could potentially enhance overall power production further. For instance, Korb et al. (2023) explored a three-turbine setup with the two upstream turbines employing Helix and the downstream turbine employing baseline control. They found that the power output depends on the phase shift between the two helix wakes, though they did not propose a method for achieving this phase difference. Similar
results were found in a three-turbine wind tunnel experiment with DIC (van Vondelen et al., 2024c). However, both studies also observe power losses at certain phase shifts, highlighting the importance of optimizing the synchronization.

While Korb et al. (2023) and van Vondelen et al. (2024c) have shown that phase differences between periodic wakes in a multi-turbine setup can influence power production, still, no robust methods have been proposed to synchronize turbines based on the phase of the upstream wake. Expanding to deeper arrays is critical to fully realizing the potential of advanced wake
control strategies in optimizing wind farm performance. Hence, it is critical to address this gap.

To establish the concept of synchronized Helix wake mixing, van Vondelen et al. (2023) suggested synchronizing turbines by estimating the phase of the incoming Helix wake on the downstream turbine from the blade loads using a linear Kalman filter, allowing for downstream control actions that incorporate the Helix's phase and any desired phase offset. While promising, this technique may only provide accurate phase estimates near the system model's linearization point, and phase estimates may
deteriorate when the turbine's state goes far from this region. Recognizing its potential, the approach has been patented despite the need for further improvements (van Vondelen et al., 2023b).

A more versatile approach is proposed in van Vondelen et al. (2024b), where an output feedback controller is designed for the downstream turbine to maintain a magnitude reference on the periodic load caused by the impingement of the Helix wake, essentially amplifying the Helix while preserving phase. This method allows for robust amplitude control of the periodic wake
and thus gives direct control of the magnitude of the load. Being output-only, it cannot apply an out-of-phase control action, as the blade is both the sensor and the actuator. It is not possible to discern the phase of the load effect of the incoming wake from the total load, which also contains the effect of an out-of-phase control action. This in-phase synchronization approach has demonstrated a 6% power improvement on the third turbine, beyond the power increase of the baseline Helix effect. However, according to Korb et al. (2023), better performance may be achieved with an out-of-phase shift, suggesting further potential
for power gains through out-of-phase synchronization.





As such, this study proposes an Extended Kalman Filter (EKF)-based phase synchronization method, building on the initial concept from van Vondelen et al. (2023). The novel controller employs an EKF and is capable of handling nonlinearities addressing the limitations of the linear Kalman filter used in van Vondelen et al. (2023). Also, it utilizes a dynamic blade-element momentum (dynBEM) model in the EKF as suggested by Coquelet et al. (2024b), which should provide a more robust
model of the simulated turbine while performing dynamic pitch actions compared to regular BEM. The main contributions of this study are as follows:

1. **Extending the EKF-based estimator for Helix phase detection:** We extend the EKF-based wind speed estimator incorporating dynBEM (Coquelet et al., 2024b) to track Helix flow oscillations regardless of dynamic pitching. By inclusion of a parametric model, including several coordinate transformations, and tuning of the internal models, the
estimator is able to isolate the wake's phase from blade load signals—thus enabling synchronized Helix control.

2. **Development of a synchronized Helix wake mixing control framework:** The EKF-based estimator for Helix phase detection is integrated into a closed-loop control framework for synchronized Helix wake mixing control on downstream wind turbines.

3. **Application to multi-turbine Helix control:** The proposed synchronized Helix wake mixing method is applied to a
three-turbine array, demonstrating significant power gains while systematically exploring the impact of phase offsets on power production and structural loads.

4. **Comprehensive validation framework:** High-fidelity large-eddy simulations coupled with OpenFAST provide a detailed validation of the proposed method, offering insights into optimal configurations and real-world applicability.

5. **Insights into wake dynamics and interference mechanisms:** The study provides novel insights into how constructive
and destructive wake interference influences wake recovery by examining wake centerlines and velocity deficits across the simulation domain.

The remainder of this paper is organized as follows. Section 2 introduces the estimation and control methods. Section 3 presents the high-fidelity simulation framework used in this work, and Section 4 presents the results. A discussion is provided in Section 5, and the paper is concluded in Section 6.

**2   Estimation and Control Methods**

This section presents the estimation and control methods that form the foundation of the proposed phase synchronization method. It covers the fundamentals of the Helix approach for wake mixing control and introduces the Extended Kalman Filter as the tool for estimating the wake phase shift and enabling synchronized control actions. Furthermore, the parametrization of the Helix wake required for detection is provided, along with the phase synchronization controller design. Lastly, specific
tuning methodology is discussed, for both the dynBEM and estimator.





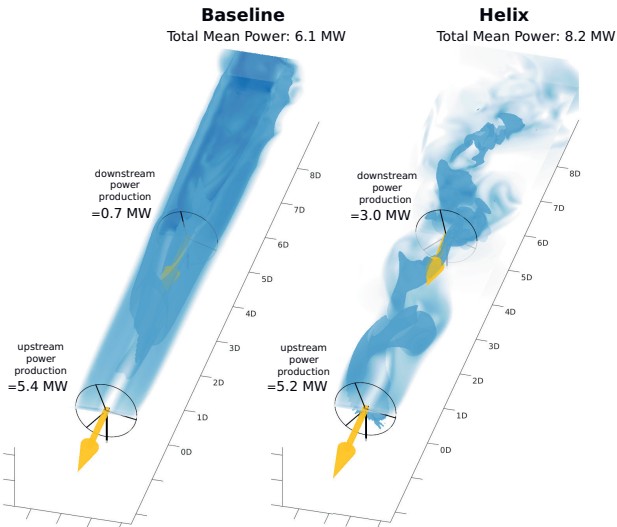

**Figure 1.** Comparison of the Helix approach (right) and the baseline case (left) in a two-turbine setup during full wake overlap, based on data from a Large-eddy simulation study in purely laminar inflow by (Frederik et al., 2020a). The image, adapted from (Meyers et al., 2022), shows the velocity magnitude in light blue and the isosurface of velocity in dark blue. The $x$-axis indicates the turbine spacing normalized by the rotor diameter $D$.

## 2.1 The Helix Approach

The Helix approach is an open-loop control strategy that enhances the power output of downstream wind turbines by applying periodic signals to the upstream wind turbine's blades (see Fig. 1 for an illustration).

Typically, the actuation commands are provided as *tilt* and *yaw* commands in the 'fixed' coordinate frame. Using the so-called backward Multi-Blade Coordinate (MBC) transformation, the Helix tilt and yaw commands ($\beta_\mathrm{tilt}$ and $\beta_\mathrm{yaw}$) are converted into effective pitch commands for each blade ($\beta_i$, where $i = 1, 2, 3$):

$$\begin{bmatrix} \beta_1 \\ \beta_2 \\ \beta_3 \end{bmatrix} = \underbrace{\begin{bmatrix} 1 & \cos{(\psi_1 + \psi_\mathrm{o})} & \sin{(\psi_1 + \psi_\mathrm{o})} \\ 1 & \cos{(\psi_2 + \psi_\mathrm{o})} & \sin{(\psi_2 + \psi_\mathrm{o})} \\ 1 & \cos{(\psi_3 + \psi_\mathrm{o})} & \sin{(\psi_3 + \psi_\mathrm{o})} \end{bmatrix}}_{T_\mathrm{cm}^{-1}(\psi(t) + \psi_\mathrm{o})} \begin{bmatrix} \beta_\mathrm{col} \\ \beta_\mathrm{tilt} \\ \beta_\mathrm{yaw} \end{bmatrix}, \tag{1}$$

where $\psi_i$ is the azimuthal position of blade $i$ (see Fig. 2 for definition) and $\psi_\mathrm{o}$ is an azimuth offset accounting for unmodeled actuator delays and blade flexibility, which is required to fully decouple the tilt and yaw channels (Mulders et al., 2019; van Vondelen et al., 2024a). The pitch angle sign convention is the following: increasing the pitch angle corresponds to pitching to feather and reduces the force intensity on the blade while decreasing the pitch angle value corresponds to pitching to stall and increases the force intensity on the blade. The collective pitch, $\beta_\mathrm{col}$, is excluded hereafter as it is regulated by the collective pitch



controller, which adjusts the pitch angle of all blades in response to rotor speed feedback, optimizing wind turbine performance by maintaining consistent power output and rotor speed.

Although multi-sine approaches have also been explored (see Huang et al. (2023)), here we consider the pure sine approach, so the Helix control commands for tilt and yaw are defined as follows:

$$
\begin{bmatrix} \beta_{\text{tilt}} \\ \beta_{\text{yaw}} \end{bmatrix} = \begin{bmatrix} A\sin(\omega_{\text{e}} t) \\ A\sin(\omega_{\text{e}} t \pm \pi/2) \end{bmatrix},
\tag{2}
$$

The excitation frequency $\omega_{\text{e}} = 2\pi f_{\text{e}}$ is governed by the dimensionless Strouhal number, $St$, calculated as:

$$
St = \frac{f_{\text{e}} D}{U_\infty},
\tag{3}
$$

where $D$ is the rotor diameter, and $U_\infty$ denotes the free-stream wind velocity. Previous studies recommend Strouhal numbers between 0.2 and 0.4 for optimal performance (Goit and Meyers, 2015; Frederik et al., 2020a). The amplitude $A$ is typically limited to a maximum of a few degrees due to practical constraints like pitch rate limitations.

The out-of-plane bending moments ($M_1$, $M_2$, $M_3$) can similarly be analyzed in the fixed frame. Here we use the forward MBC transformation to obtain $M_{\text{col}}, M_{\text{tilt}}, M_{\text{yaw}}$:

$$
\quad \begin{bmatrix} M_{\text{col}} \\ M_{\text{tilt}} \\ M_{\text{yaw}} \end{bmatrix} = \frac{2}{3} \underbrace{\begin{bmatrix} 1/2 & 1/2 & 1/2 \\ \cos(\psi_1) & \cos(\psi_2) & \cos(\psi_3) \\ \sin(\psi_1) & \sin(\psi_2) & \sin(\psi_3) \end{bmatrix}}_{T_{\text{cm}}(\psi(t))} \begin{bmatrix} M_1 \\ M_2 \\ M_3 \end{bmatrix}.
\tag{4}
$$

The sign convention is provided in Fig 2: positive tilt moment corresponds to an overload on the top part of the rotor, while positive yaw moment corresponds to an overload on the right part of the rotor.

Note that (1) and (4) can be used to go back and forth between the fixed and rotating domain for any blade-effective signal.

There are two variants of the Helix approach: clockwise (CW) and counter-clockwise (CCW) rotation. The CW variant is
implemented by setting $-\pi/2$ in $\beta_{\text{yaw}}$, while the CCW variant uses $+\pi/2$. Although both variants maintain the same actuation frequency in the fixed frame, the effective frequency experienced by the pitch actuator differs when these commands are translated to the rotating frame, yielding:

$$
\beta_i = \beta_{\text{col}} + \cos(\psi_i + \psi_{\text{o}})\beta_{\text{tilt}} + \sin(\psi_i + \psi_{\text{o}})\beta_{\text{yaw}},
\tag{5}
$$

which leads to the Helix frequency in the rotating frame being the sum or difference of the rotor's rotational frequency $\omega_r$ and
the excitation frequency $\omega_e$, depending on whether the rotation is CCW or CW, respectively:





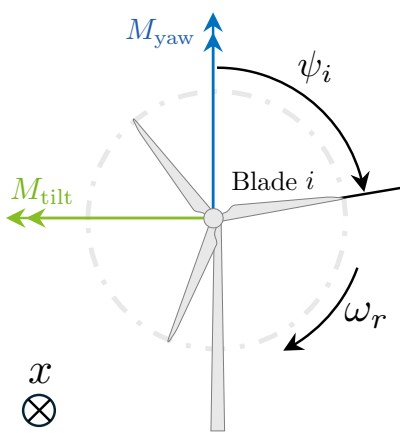

**Figure 2.** Graphic representation of blade azimuth $\psi_i$ and associated sign convention for fixed-frame moments $M_{\text{tilt}}$ and $M_{\text{yaw}}$.

$$\beta_i = \beta_{\text{col}} + \cos(\omega_r t + \psi_i^0 + \psi_{\text{o}})\beta_{\text{tilt}} \tag{6}$$
$$+ \sin(\omega_r t + \psi_i^0 + \psi_{\text{o}})\beta_{\text{yaw}},$$
$$= A\cos(\omega_r t + \psi_i^0 + \psi_{\text{o}})\sin(\omega_e t)$$
$$+ A\sin(\omega_r t + \psi_i^0 + \psi_{\text{o}})\sin(\omega_e t \pm \pi/2)$$
$$= A\sin[(\omega_r \pm \omega_e)t + \psi_i^0 + \psi_{\text{o}}], \tag{7}$$

where $\psi_i^0$ is the azimuthal position of blade $i = 1, 2, 3$ at $t = 0$. Typically, the CCW Helix variant yields greater energy gains for the downstream turbine (Frederik et al., 2020a; Taschner et al., 2023), while the CW Helix is preferred for its reduced impact on pitch bearing wear (van Vondelen et al., 2023a), attributed to the lower effective actuation frequency of $\omega_r - \omega_e$.

The employment of the Helix approach induces periodic loading, affecting the fatigue life of the turbine performing the actuation (van Vondelen et al., 2023a). This periodic loading also extends to downstream turbines, as observed by Frederik and van Wingerden (2022), which directly results from interaction with the periodic structure in the wake induced by the upstream turbine (see Fig. 1). Manipulating this periodic structure in the wake by actuating the Helix approach on the downstream turbine in an in-phase/out-of-phase synchronized fashion could potentially enhance wake mixing and, thereby, power production downstream even more (van Vondelen et al., 2024b; Korb et al., 2023). Phase estimation in Helix wake mixing can be understood as an implicit wind speed estimation, as the wake's periodic structure is influenced by variations in incoming flow. To achieve this, we estimate the incoming wind speed and extract phase information of the periodic wind speed component using an estimator. One way to estimate this phase shift is through state estimation techniques, such as the extended Kalman filter, which we introduce below.



## 2.2 Estimation using the Extended Kalman Filter

The EKF is an extension of the Kalman Filter and is tailored for nonlinear systems. It approximates the nonlinear state and measurement models through linearization, enabling state estimation. The EKF is widely used to estimate states in systems where the relationships between variables are nonlinear, providing a means to manage the associated uncertainties effectively. This section describes how the EKF is leveraged in this work (see e.g. Chui and Chen (2017) for extensive EKF theory). First, we have a nonlinear system description:

$$\mathbf{x}_{k+1} = f(\mathbf{x}_k, \mathbf{u}_k) + \mathbf{w}_k, \tag{8}$$

$$\mathbf{y}_k = h(\mathbf{x}_k, \mathbf{u}_k) + \mathbf{v}_k, \tag{9}$$

where $\mathbf{x}_k$ is the state vector, $\mathbf{u}_k$ is the control input vector, $\mathbf{y}_k$ is the measurement vector, $\mathbf{w}_k$ is the process noise vector, and $\mathbf{v}_k$ is the measurement noise vector. The functions $f(\cdot)$ and $h(\cdot)$ are the state and measurement functions, respectively. Let us now tailor this general representation for our application. A distinction can be made between controllable and uncontrollable

inputs:

$$\mathbf{u}_k = \begin{bmatrix} \mathbf{u}_k^{\mathrm{c}} \\ \mathbf{u}_k^{\mathrm{u}} \end{bmatrix}, \tag{10}$$

where $\mathbf{u}_k^{\mathrm{c}}$ is the input vector containing *controllable* inputs, which are the pitch control inputs in our case, and $\mathbf{u}_k^{\mathrm{u}}$ is the input vector containing *uncontrollable* inputs, which is the incoming periodic Helix wake impinging on the downstream turbine. The goal is to estimate the unknown uncontrollable input vector $\mathbf{u}_k^{\mathrm{u}}$ by treating it as the state to be estimated. This can be achieved

by assuming the following model representation (see e.g. Verhaegen and Verdult (2007)):

$$\mathbf{u}_{k+1}^{\mathrm{u}} = \mathbf{u}_k^{\mathrm{u}} + \mathbf{w}_k^{\mathrm{u}}, \tag{11}$$

commonly known as the random walk model. Note that this model can only be assumed for biases or slowly varying states, which is a reasonable assumption in our application as we estimate constant parameters that define a sinusoid (treated in Section 2.3). In the case of a periodic state, an undamped oscillator may be a more suitable model (van Vondelen et al., 2023). The

state-space system becomes:

$$\underbrace{\begin{bmatrix} \mathbf{x}_{k+1} \\ \mathbf{u}_{k+1}^{\mathrm{u}} \end{bmatrix}}_{\mathbf{x}_{k+1}^{\mathrm{aug}}} = \begin{bmatrix} f(\mathbf{x}_k, \mathbf{u}_k^{\mathrm{u}}, \mathbf{u}_k^{\mathrm{c}}) + \mathbf{w}_k \\ \mathbf{u}_k^{\mathrm{u}} + \mathbf{w}_k^{\mathrm{u}} \end{bmatrix}, \tag{12}$$

$$\mathbf{y}_k = h(\mathbf{u}_k^{\mathrm{u}}, \mathbf{u}_k^{\mathrm{c}}) + \mathbf{v}_k. \tag{13}$$

Using an EKF, it is now possible to estimate the uncontrollable input vector $\mathbf{u}_k^{\mathrm{u}}$. The measurement function $h(\mathbf{u}_k^{\mathrm{u}}, \mathbf{u}_k^{\mathrm{c}})$ is chosen to be the dynBEM model which computes the blade out-of-plane loads based on blade-effective wind speeds (BEWS), rotor

velocity, and pitch angle in the rotating (blade) coordinate frame. It represents a nonlinear measurement mapping dependent on



both the controllable and uncontrollable inputs. Note that the actual system state equation is unknown, and dynBEM does not depend on it; the dynamics are modeled according to the engineering model of Snel and Schepers (1995). Since we are only interested in estimating $\mathbf{u}_k^{\mathrm{u}}$, we can formulate the EKF problem as follows:

$$\hat{\mathbf{u}}_{k+1}^{\mathrm{u}} = \hat{\mathbf{u}}_k^{\mathrm{u}} + K_k \mathbf{e}_k, \tag{14}$$

$$\mathbf{e}_k = \mathbf{y}_k - h(\hat{\mathbf{u}}_k^{\mathrm{u}}, \mathbf{u}_k^{\mathrm{c}}), \tag{15}$$

where $\hat{(\cdot)}$ denotes an estimate, $K_k$ is the Kalman gain, and $\mathbf{e}_k$ is the innovation signal vector.

BEWS represents the local wind speed experienced by each blade as it rotates through the wake-affected flow field. Unlike freestream wind speed, BEWS accounts for variations due to wake dynamics, turbulence, and aerodynamic interactions.

Next to the above-mentioned turbine signals, dynBEM requires system parameters such as the rotor radius, hub radius, and
airfoils (see Coquelet et al. (2024b) for additional details on using dynBEM in a wind speed estimator).

To calculate the Kalman gain, it is assumed that estimates of the covariance matrices of $\mathbf{w}_k$ and $\mathbf{v}_k$ are available:

$$E\left[\begin{bmatrix} \mathbf{w}_k \\ \mathbf{v}_k \end{bmatrix} \begin{bmatrix} \mathbf{w}_k^T & \mathbf{v}_k^T \end{bmatrix}\right] = \begin{bmatrix} Q & S^T \\ S & R \end{bmatrix} \succeq 0, \tag{16}$$

where $Q$ and $R$ are the covariance matrices of $\mathbf{w}_k$ and $\mathbf{v}_k$, respectively, $S$ is their cross-covariance, and the matrix is assumed to be positive semi-definite.

The state transition matrix $F_k$ is the identity matrix based on the definition of the state equation (Eq. (14)), simplifying the covariance propagation. Additionally, the process noise $\mathbf{w}_k$ and measurement noise $\mathbf{v}_k$ are assumed to be uncorrelated, allowing the cross-covariance term $S$ to be neglected. This assumption is valid given the independent nature of the noise sources in the system.

The Kalman gain is then obtained by first propagating the Riccati difference equation:

$$P_{k+1} = P_k + Q - K_k P_k H_k^{\mathrm{u}T}, \tag{17}$$

where $P_k$ is the covariance matrix estimate, and $H_k^{\mathrm{u}}$ is the Jacobian of the measurement function $h$ with respect to $\mathbf{u}^{\mathrm{u}}$.

Note that we do not have a differentiable nonlinear expression for the measurement function (Eq. (13)). As such, the Jacobian $H_k^{\mathrm{u}}$ is approximated by central differences, given that $h(\cdot)$ is a nonlinear function:

$$H_k^{\mathrm{u}} \approx \frac{h(\hat{\mathbf{u}}_k^{\mathrm{u}} + \mathrm{d}n/2, \mathbf{u}_k^{\mathrm{c}}) - h(\hat{\mathbf{u}}_k^{\mathrm{u}} - \mathrm{d}n/2, \mathbf{u}_k^{\mathrm{c}})}{\mathrm{d}n}, \tag{18}$$

where $\mathrm{d}n$ is a small deviation from the operating point. The choice of $\mathrm{d}n$ requires balancing truncation and round-off errors. Typically, $\mathrm{d}n$ should be small relative to $u_k$, often in the range of $10^{-5}$ to $10^{-2}$ times $u_k$, ensuring numerical accuracy without excessive sensitivity to floating-point precision.

Finally, the Kalman gain is calculated as:

$$K_k = P_k H_k^{\mathrm{u}T} (R + H_k^{\mathrm{u}} P_k H_k^{\mathrm{u}T})^{-1}. \tag{19}$$





The Kalman gain $K_k$ and error covariance $P_k$ generally converge to steady-state values under constant wind conditions. However, in realistic wind farm environments, wind speed variations and turbulence influence the uncertainty in phase estimation, requiring $K_k$ to adapt dynamically. The EKF inherently provides some adaptivity by updating state estimates in real time, but further improvements can be achieved by tuning the process noise $Q$ and measurement noise $R$ based on observed wind conditions, which is discussed in Section 2.5. The next section presents the parameterization of the Helix wake used in 205   the EKF.

## 2.3   State vector: Helix wake representation

As described in Section 2.1, the Helix wake is generated by periodic control actions applied to an upstream turbine (T1). While the fluid dynamics phenomena relating the actuation and the shape of the wake remain an active area of research (Coquelet et al., 2024a; Korb et al., 2023), a consensus is that the Helix wake propagates following a helical shape (see Fig. 1). From 210   a downstream turbine (T2) perspective, this implies that the incoming wind field consists of a wake deficit that is misaligned from the rotor center and rotates around it over time. This can be modeled as wind speed changes in the tilt and yaw directions. To capture this behavior, the wake is then represented as a $U_{\text{tilt}}$ and $U_{\text{yaw}}$ sine-like perturbation. The frequency of these perturbations is assumed equal to that of the periodic control input applied at the upstream turbine.

     This assumption on the frequency content is supported by Fig 3, which shows $U_{\text{col}}$, $U_{\text{tilt}}$ and $U_{\text{yaw}}$ extracted from the velocity 215   field at 5D behind a single turbine for a Helix and a baseline case. How $U_{\text{tilt}}$ and $U_{\text{yaw}}$ are retrieved from the slice of velocity is presented in Section 3.3.1. A clear peak at the Helix frequency can be observed, further suggesting that the Helix-induced wake structure, while subject to turbulent mixing, retains a coherent oscillatory pattern as it travels through the wind farm (van Vondelen et al., 2024b; Frederik and van Wingerden, 2022).

     The model for the wind speed $U$ at a downstream turbine T2 is then expressed as:

$$
\begin{bmatrix} U_{\text{col}} \\ U_{\text{tilt}} \\ U_{\text{yaw}} \end{bmatrix} = \begin{bmatrix} A_{\text{col}} \\ A_{\text{helix}} \sin(\omega_{\text{e}} t + \varphi_{\text{tilt}}) \\ A_{\text{helix}} \cos(\omega_{\text{e}} t + \varphi_{\text{yaw}}) \end{bmatrix},
\tag{20}
$$

where $A_{\text{col}}$ represents the collective wind speed component, and $A_{\text{helix}}$ is the amplitude of the periodic components of the Helix wake. The collective amplitude $A_{\text{col}}$ therefore represents the mean wind speed over the rotor swept area. The sign convention and physical meaning of $U_{\text{tilt}}$ and $U_{\text{yaw}}$ are similar to those of the rotor moments: a positive $U_{\text{tilt}}$ corresponds to an over-speed on the top part of the rotor (naturally causing a positive tilt moment if no individual pitch control (IPC) command is applied on 225   T2), while positive $U_{\text{yaw}}$ corresponds to an over-speed in the right part of the rotor (naturally causing a positive tilt moment if no IPC command is applied on T2). The parameters $\varphi_{\text{tilt}}$ and $\varphi_{\text{yaw}}$ represent the phase shifts between the tilt and yaw actuation on T1 and the tilt and yaw wind speed perturbation of the wake impinging on T2 (see Fig. 6). If the wake propagated as a perfect helix, these parameters would be identical. As distortion happens as the wake propagates downstream, an additional degree of freedom is given to the model by considering distinct values for these offsets in the tilt and yaw direction. These phase shifts 230   are the main parameters of interest as they influence the alignment of the wake structures with downstream turbines.

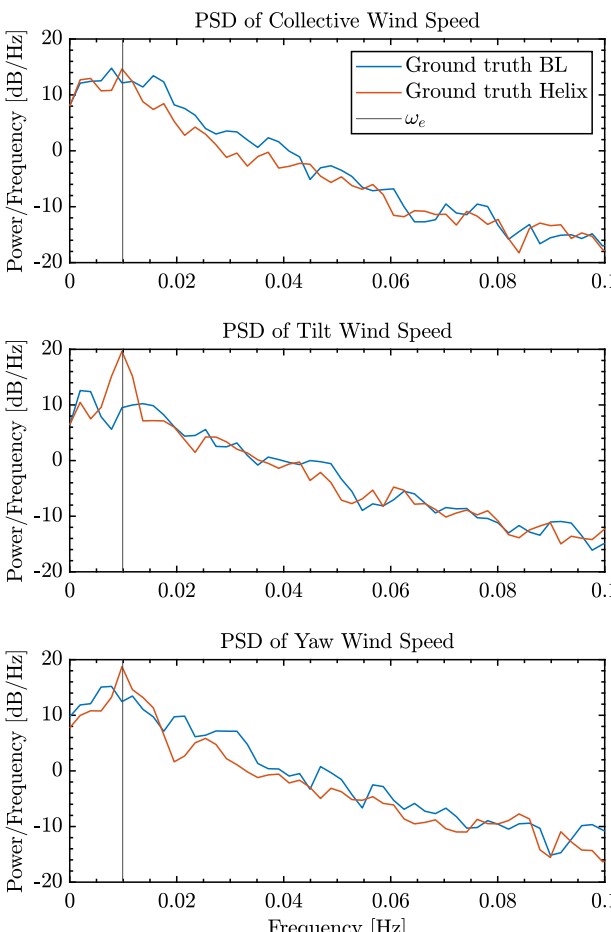

**Figure 3.** Power spectral density comparison at 5D downstream from a turbine using Baseline control (blue) and Helix control (red). The Helix wake shows a distinct peak at the excitation frequency $f_e$ (black line), confirming the periodic nature of the wake. This periodicity is required for phase estimation and synchronization in downstream turbines.





The purpose of the estimator is then to estimate the parameters of the model described by Eq. 20, and the following state vector is employed:

$$\mathbf{u}_k^u = \begin{bmatrix} A_{\text{col}} \\ A_{\text{helix}} \\ \varphi_{\text{tilt}} \\ \varphi_{\text{yaw}} \end{bmatrix}. \tag{21}$$

This state vector captures both the amplitude and phase information necessary to describe the wind speed components at the downstream turbine. Given that these parameters are typically constant under steady operating conditions, they are modeled as random walks in the estimation process. This approach allows for capturing slow variations in the wake characteristics due to changing environmental conditions or turbine dynamics.

By directly estimating the phase shifts $\hat{\varphi}_{\text{tilt}}$ and $\hat{\varphi}_{\text{yaw}}$ from the observed wind speed data, the model provides the necessary information for downstream synchronization control. The next section presents how we tune the EKF internal model, i.e. the measurement function.

### 2.4 EKF internal model specifications and tuning

In the estimation procedure, the state vector $\mathbf{u}_k^u$ is used by the EKF internal model through the measurement function $h$ (see Eq. 13). As mentioned in Section 2.2, this work relies on the BEM theory (Coquelet et al., 2024b), which computes the blade out-of-plane loads based on BEWS, rotor velocity, and pitch angle in the rotating (blade) coordinate frame. The state vector therefore needs to be adapted to fit the required inputs of the BEM, i.e. BEWS. The state vector is used in Eq. 20 to provide the vector of wind speed perturbation in the rotor frame, consisting of $U_{\text{col}}$, $U_{\text{tilt}}$ and $U_{\text{yaw}}$. This vector is then mapped onto the rotating frame using the backward MBC transform defined in Eq. (1), eventually leading to BEWS that are usable by the BEM model (see Fig. 7 for the illustration of the flow of information in the estimation process).

Wind turbine models like the BEM can capture the essential dynamics of the system; however, delays can still occur due to factors like actuator response time, induced velocities reaction time, or blade flexibility. To treat these delays, this work combines two approaches given in the literature.

On the first hand, the pitch response delays are accounted for using an optimal azimuth offset to be used in the backward MBC transform (Eq. 1) (Mulders et al., 2019). This ensures that the pitch angle fed to BEM is the actual blade pitch angle and not the pitch command. Identifying the optimal azimuthal offset is typically done using the relative gain array (RGA) of a linearized system model, and can depend on the simulation tool, as the delays reflect unmodeled dynamics. Van Vondelen et al. (2024a) proposed a method for directly identifying the optimal azimuth offset within the simulation environment through system identification. Using that methodology, we identified the optimal azimuth offset of $\phi_{\text{off}} = 17°$ for our scenario. Note that this offset is highly model-dependent and should be determined for each specific wind turbine model and/or configuration.

On the other hand, we account for the aerodynamic delays appearing between the pitch angle changes and the blade forces response using a dynamic version of the BEM. It was indeed shown in Coquelet et al. (2024b) that dynamic effects appear in





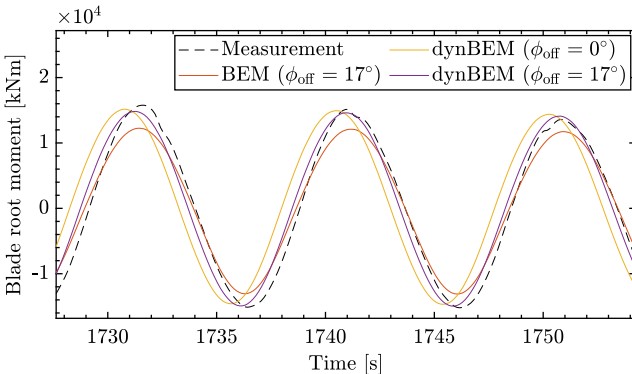

**Figure 4.** Blade root moments comparison with and without optimal azimuth offset $\phi_{\mathrm{off}}$ using the static BEM and dynamic BEM models, highlighting the improved match of the dynBEM model with azimuth offset to LES data.

the local inductions computed at the blade due to the Helix pitch. Those are not modeled by the standard BEM model as the BEM theory considers steady operation.

Figure 4 demonstrates the effect of these two corrections of this optimal azimuth offset on the blade root moments computed using the BEM and compared to a reference LES consisting of an upstream turbine operating the Helix in a laminar inflow.

The figure compares:

- the standard BEM output when the pitch values are the actual pitch angles (Eq. 1 with optimal azimuth offset),

- the dynBEM output when the pitch values are identical to the pitch commands (Eq. 1 with no azimuth offset),

- the dynBEM output when the pitch values are the actual pitch angles following the actuator response (Eq. 1 with optimal azimuth offset).

The figure shows that incorporating the optimal azimuth offset significantly improves the alignment between the model output and actual measurements. Additionally, the dynBEM model, which can be tuned with a single parameter, shows a closer match to the LES data compared to the static BEM model, indicating its superior ability to capture dynamic effects. The next section describes the tuning of the noise covariance matrices.

## 2.5 EKF noise parameters tuning

The EKF relies on accurately defined process noise (Q) and measurement noise (R) covariance matrices, which are often challenging to estimate accurately. This section presents methodologies for tuning these matrices, starting with estimating the R matrix.

The process begins by selecting a frequency band of interest for the system's dynamics, treating frequencies outside this band as noise components (see Fig. 5). We employ a high-pass filter on the signal to isolate this noise, extracting the high-frequency
components. The variance of this filtered output is then adopted as the entries of the R matrix.



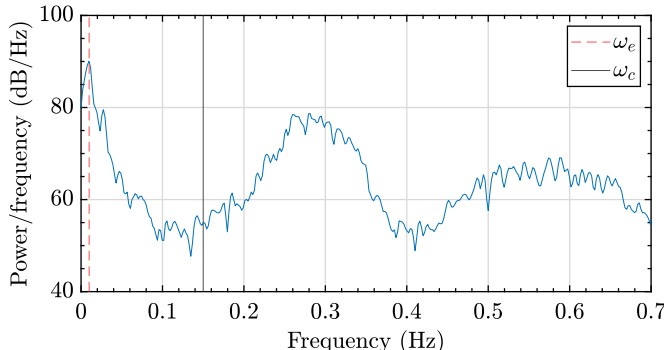

**Figure 5.** Power spectrum analysis of $M_{\text{tilt}}$: Determining the measurement noise covariance matrix (R). The Helix frequency is highlighted with a red dashed line, and the cut-off frequency $\omega_c$ is denoted by the black line.

The estimation of the Q matrix involves more nuanced steps. Initially, the Helix wind speed component is determined from the ground truth wind speed data (details on the ground truth provided in Sections 3.3 and 4.1) using a low-pass filter. This ensures that only the frequencies contributing significantly to the signal's behavior are considered. After filtering, the amplitude and phase shift of the wind speed are computed, forming the basis for the Q matrix selection. Specifically, the Q matrix is set as the variance of the derivative of these filtered signals, providing a measure of process noise. In real-world conditions, actual wind speed measurements and similar post-processing could provide an initial value for the Q matrix.

After the initial estimates, the Q matrix requires further refinement to enhance the EKF's performance. This is achieved through iterative tuning, where the matrix is scaled by a constant factor based on the observed performance of the filter in simulation trials. This iterative process continues until the EKF's performance aligns with the desired accuracy and reliability criteria. The pseudo-code in Alg. 1 summarizes the tuning process.

To enhance the accuracy and robustness of the proposed EKF-based synchronization method, real-time calibration can be a suitable addition. This approach involves dynamically adjusting the estimator's parameters to respond to changing environmental conditions and operational dynamics (Mehra, 1970). One critical aspect of real-time calibration is the adaptive adjustment of the process noise (Q) and measurement noise (R) covariance matrices within the EKF. These matrices, initially set during offline tuning according to Alg. 1, may not fully capture the complexities of real-world conditions, where wind speeds, turbulence, and atmospheric stability can vary significantly. By continuously monitoring the estimator's performance, particularly through the innovation sequence (the difference between predicted and actual measurements, Eq. (15)), the system can detect when the current noise assumptions are inadequate. For instance, during periods of high turbulence, increasing the process noise covariance can account for greater uncertainty in wind speed estimates, leading to more accurate control actions. However, this is out of the scope of the current work. The next section presents the phase synchronization controller.





---

**Algorithm 1** Tuning the EKF Covariance Matrices

---

1: **Input:** Raw signal data from the wind turbine

2: **Output:** Optimized Q and R matrices for the EKF

3: **Step 1: Estimate Measurement Noise Covariance (R)**

    1. Identify the frequency band of interest in the signal.

    2. Apply a high-pass filter to isolate high-frequency noise.

    3. Compute the variance of the filtered signal.

    4. Set the R matrix to this computed variance.

4: **Step 2: Estimate Process Noise Covariance (Q)**

    1. Apply a low-pass filter to the ground truth wind speed signal to isolate the Helix component.

    2. Calculate the amplitude and phase shift of the filtered signal.

    3. Compute the variance of the derivative of these filtered signals.

    4. Set the Q matrix to this computed variance.

5: **Step 3: Fine-Tune Q Matrix**

    1. Initialize the EKF with the estimated Q matrix.

    2. **while** EKF performance is unsatisfactory **do**

        (a) Scale the Q matrix by a constant factor.

        (b) Re-run the EKF and assess performance against validation data.

        (c) Adjust the scaling factor as necessary.

    3. **end while**

---

## 2.6 Phase synchronization controller design

The synchronization controller is designed to align the control actions of downstream turbines with the phase of the incoming wake generated by upstream turbines. It utilizes phase estimates provided by the EKF, which tracks the phase shifts $\hat{\varphi}_{\text{tilt}}$ and $\hat{\varphi}_{\text{yaw}}$ of the incoming wake as described in the previous section. Based on these estimates, the controller adjusts the tilt and yaw control signals of the downstream turbines. The control signals are expressed as:

$$\begin{bmatrix} \beta_{\text{tilt}} \\ \beta_{\text{yaw}} \end{bmatrix} = \begin{bmatrix} A\sin(\omega_{\text{e}}t + \hat{\varphi}_{\text{tilt}} + \varphi_{\text{off}}) \\ A\sin(\omega_{\text{e}}t \pm \pi/2 + \hat{\varphi}_{\text{yaw}} + \varphi_{\text{off}}) \end{bmatrix}, \tag{22}$$



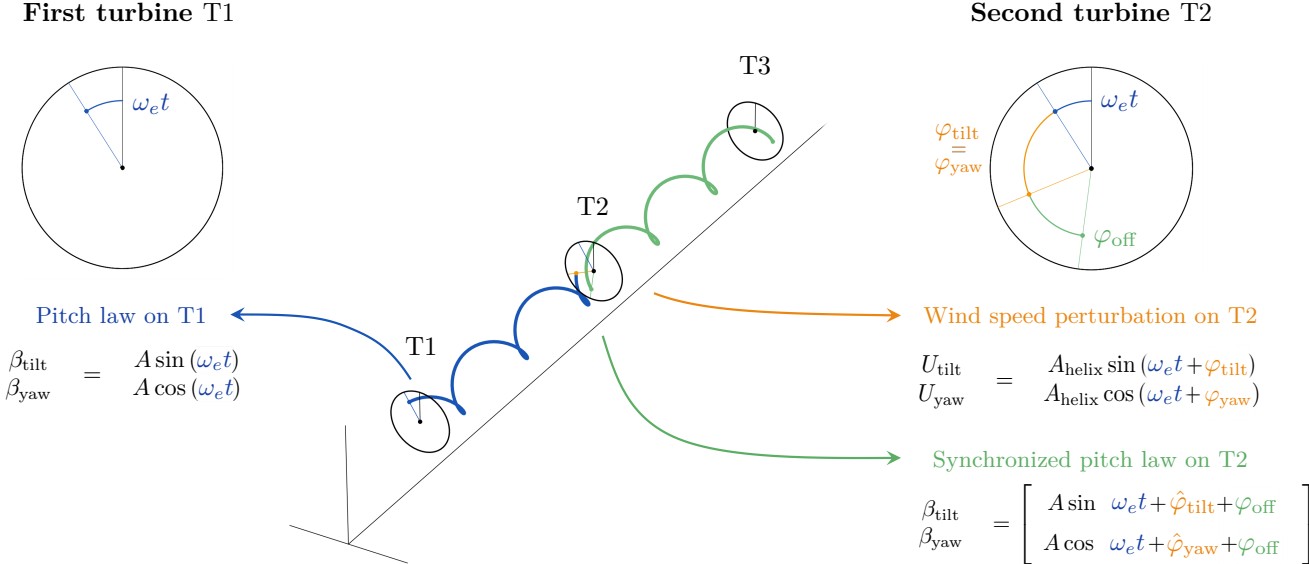

**Figure 6.** Graphical representation of the phase synchronization controller. The control action on T2 is based on the estimated phase shift of the incoming Helix wake $\hat{\varphi}_{\text{tilt}}$ and $\hat{\varphi}_{\text{yaw}}$, and a user-defined phase offset $\varphi_{\text{off}}$. Note that in this figure the CCW Helix is illustrated.

where $A$ is the amplitude, $\omega_e$ is the excitation frequency, and $\varphi_{\text{off}}$ is an additional phase shift that can be applied to modify the alignment of the up-and downstream Helix wakes. A graphical representation of the synchronization is provided in Fig. 6. The phase offset is defined such that

– $\varphi_{\text{off}} = 0$ indicates that the pitch is in phase with the wind speed perturbation (see Eq. 20): the blade pitch increases as a blade travels through the area of higher wind speed, hence the blade forces are locally reduced compared to a case where IPC is not active;

– $\varphi_{\text{off}} = 180°$ indicates that the pitch is out-of-phase with the wind speed perturbation (see Eq. 20): the blade pitch decreases as a blade travels through the area of higher wind speed, hence the blade forces are locally increased compared

to a case where IPC is not active.

The adjusted control signals are then implemented by the downstream turbine's pitch system (Eq. (1)), which alters the rotor's tilt and yaw to create a controlled wake. The overall process is focused on achieving a precise phase offset between the downstream turbine's control actions and the incoming wake, using the EKF-derived phase information to inform these adjustments. A schematic of the entire framework, including the proposed estimator and control strategy is given in Fig. 7. The

next section will present the framework used for evaluating the proposed method.





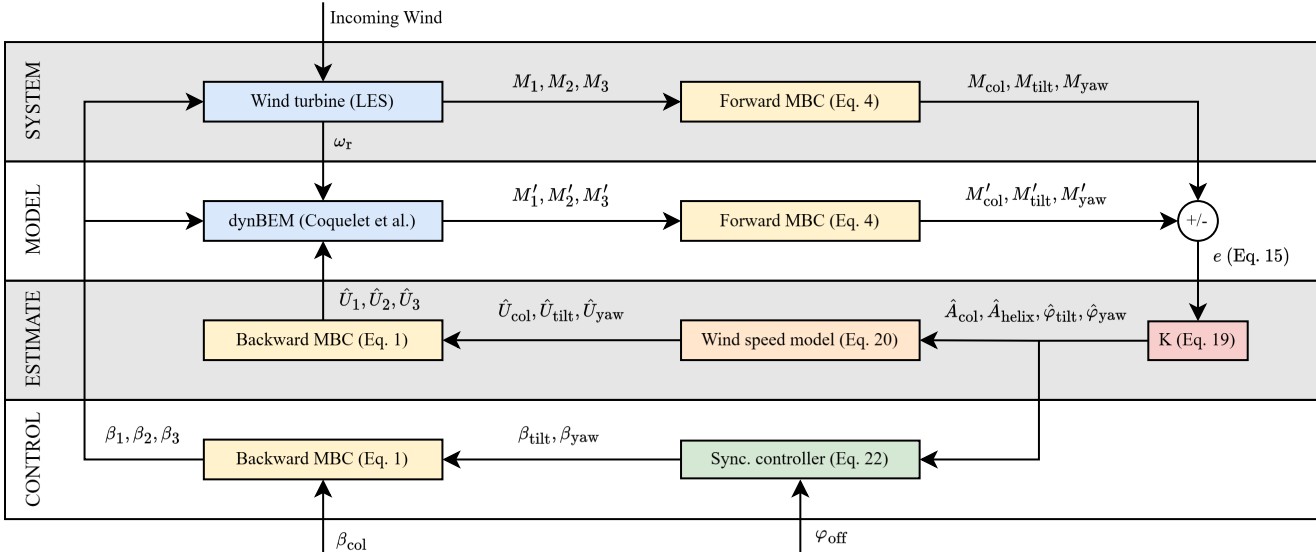

**Figure 7.** Schematic of the synchronized Helix wake mixing control framework illustrating the flow of information. The figure illustrates the integration of the wind turbine, EKF, dynBEM, and synchronization controller. Note that the model outputs are distinguished from the system outputs with an apostrophe. $U_i$, where $i = 1, 2, 3$, are the blade-effective wind speeds after transforming to the rotating reference frame.

## 3 High-fidelity simulation framework

This section outlines the simulation framework and configurations employed to evaluate the performance of the novel synchronization controller using the high-fidelity simulation tools OpenFAST (Jonkman et al., 2024) and AMR-Wind (Brazell et al., 2021). These tools are integrated to simulate complex wake interactions between wind turbines under realistic atmospheric conditions.

### 3.1 Large-eddy Simulation Environment

The Large-eddy simulations are performed with the AMR-Wind software, which is well suited for the study of wind farms in atmospheric boundary layer flows Brazell et al. (2021). The simulation employs a Convective Boundary Layer (CNBL) setup that includes Coriolis effects to replicate a stable stratified atmosphere interacting with the wind turbines. This setup is similar to the one used in Taschner et al. (2023) and identical to the one used in van Vondelen et al. (2024b).

The precursor simulation is performed with a domain of dimensions 5360 m in the $x$-direction, 3200 m in the $y$-direction, and 1600 m in the $z$-direction. An isotropic grid size of 10 m is used, complying with CNBL requirements (Wurps et al., 2020). Periodic boundary conditions allow the flow to evolve over 16 hours, achieving a quasi-stationary turbulent ABL state (Zil-



itinkevich et al., 2007), where the wind speed at hub height $U_{\mathrm{hub}}$ of the first turbine is forced to be 10.5 m/s and the turbulence
intensity $TI_{\mathrm{hub}}$ is around 3 %.

After this initial phase, $y$-$z$ planes at the inflow boundary ($x = 0$ m) are sampled for 45 minutes to serve as inflow conditions
for turbine simulations. For these simulations, the turbine blades are modeled using the Actuator Line Method (ALM) coupled
with OpenFAST.

The turbines, which are modeled by OpenFAST (see next section), are placed within the domain at coordinates ($x = 1200$ m,
$y = 1600$ m) for turbine 1 (T1), ($x = 2400$ m, $y = 1600$ m) for turbine 2 (T2), and ($x = 3600$ m, $y = 1600$ m) for turbine 3
(T3). This corresponds to a $5D$ spacing, where $D$ represents the rotor diameter (see Table 1), from the inflow and between the
turbines, and also sufficient space for wake development behind the third turbine.

To facilitate higher resolution flow analysis around the wind turbines, a mesh refinement to 5 m is implemented. This
refinement covers a static box area starting $4.5D$ upstream of the first turbine, with dimensions of 5040 m in the $x$-direction,
960 m in the $y$-direction, and 600 m in the $z$-direction.

## 3.2   Wind Turbine Simulation Tool

The turbine is modeled by the OpenFAST solver, which serves as a comprehensive multi-fidelity simulation tool by integrating
various modules focused on structural dynamics, aerodynamics, and control systems. The simulation setup for this study
employs the International Energy Agency's (IEA) 15 MW fixed-bottom reference wind turbine (specifics in Table 1) (Gaertner
et al., 2020). The proposed method is implemented in an external Python script that computes pitch control setpoints in real
time. These setpoints are then transmitted to ROSCO during runtime via ZeroMQ, a lightweight messaging library for high-
performance asynchronous communication (Hintjens, 2013).

In the LES-coupled simulation environment, the turbine blades are represented via the Actuator Line Method (ALM), where
each blade segment exerts dynamic forces on the surrounding air, thereby influencing local flow properties such as velocity
and turbulence (Sørensen and Shen, 2002). The interaction between OpenFAST's fine temporal resolution and the coarser LES
grid in AMR-Wind necessitates sophisticated interpolation techniques and phase adjustments to ensure accurate and timely
controller responses.

## 3.3   Simulation Cases

This section introduces the simulation cases designed to validate the proposed estimator and evaluate the control strategy. It
starts by detailing the methodology for obtaining the ground truth, followed by an overview of the synchronization cases,
including a summary of the simulation setups and the performance metrics used for evaluation.



**Table 1.** Specifications of the IEA 15 MW reference turbine used in the simulations.

| Characteristic | Value |
| --- | --- |
| Hub height | 150 m |
| Rotor diameter | 240 m |
| Rated power | 15 MW |
| Rated wind speed | 10.59 m/s |
| Cut-in wind speed | 3 m/s |
| Cut-out wind speed | 25 m/s |
| Min. rotor speed | 5 rpm |
| Max. rotor speed | 7.56 rpm |

### 3.3.1 Ground Truth for Estimator Validation

A ground truth is required for validating wind estimates from the proposed controller. The ground truth is generated through an LES involving only the first turbine (T1) in the array, capturing wind conditions unaffected by downstream turbines. This scenario establishes a baseline for comparing wind speed estimates derived from T2's blade loads.

Wind velocity data are sampled 5D downstream of T1, aligning with the location of T2 in synchronization cases. The sampling process involves:

- Sampling $n$ lines originating from the rotor center across the flow field located at azimuthal positions $\psi_{L,1}, \ldots, \psi_{L,n}$ (see Fig. 8);

- Averaging wind speed along these lines over time to obtain line-effective wind speeds $U_{L,1}, \ldots, U_{L,n}$;

- Mapping the line-effective wind speeds to fixed-frame to obtain $U_{\text{col}}$, $U_{\text{tilt}}$, and $U_{\text{yaw}}$ using the MBC transform generalized for $n$ lines proposed in Moens et al. (2022):

$$
\begin{bmatrix} U_{\text{col}} \\ U_{\text{tilt}} \\ U_{\text{yaw}} \end{bmatrix}^{\text{GT}} = \frac{2}{n} \begin{bmatrix} 1/2 & \ldots & 1/2 \\ \cos(\psi_{L,1}) & \ldots & \cos(\psi_{L,n}) \\ \sin(\psi_{L,1}) & \ldots & \sin(\psi_{L,n}) \end{bmatrix} \begin{bmatrix} U_{L,1} \\ \vdots \\ U_{L,n} \end{bmatrix}.
\tag{23}
$$

The number of lines is set to $n = 36$ in this case. A band-pass filter is applied to the fixed-frame wind speeds to isolate the Helix component. These processed data serve as a benchmark for validating the estimator in simulations with multiple turbines, facilitating the tuning of the EKF. Note that this bandpass filter is not applied on the signals shown in Fig. 3.

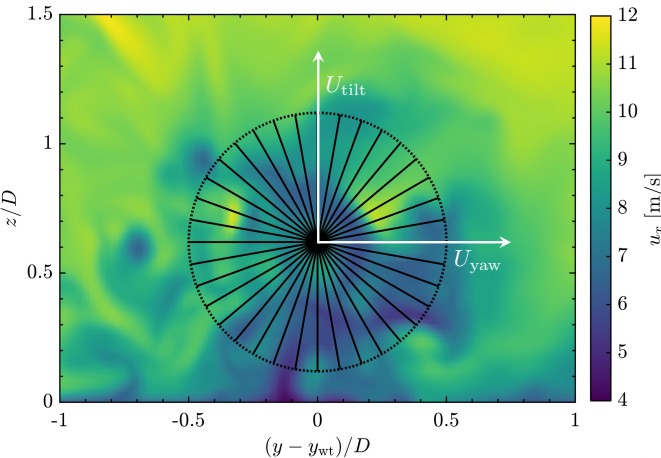

**Figure 8.** Sampling methodology for estimating blade-effective wind speeds from the LES flow field. The average wind speed on thirty-six lines is mapped to the tilt and yaw axes to derive the effective wind speed at 5D downstream in the fixed coordinate frame.

### 3.3.2 Synchronization Cases

The synchronization cases evaluate the performance of the EKF-based phase synchronization control strategy. The simulations consist of three turbines (T1, T2, and T3) arranged linearly with 5D spacing. Realistic inflow conditions are generated using a precursor LES (see Section 3).

Control strategies for the turbines are as follows:

- **T1:** Operates under Helix control in all cases.

- **T2:** Applies either baseline control or synchronized phase control depending on the scenario.

- **T3:** Always operates under baseline control to isolate the impact of upstream control strategies.

The primary objective is to explore the influence of different phase shift offsets ($\varphi_{\text{off}}$) in T2's control on downstream power production and structural loads.

### 3.3.3 Cases Summary

Ten distinct simulation cases are defined, as summarized in Table 2:

1. **T1 Only (Helix):** T1 employs Helix control, with T2 and T3 absent. This serves as the baseline for wake development and ground truth sampling (see Section 3.3.1).

2. **BL Helix:** T2 and T3 apply baseline control, while T1 operates with Helix control. This case serves as a reference for the synchronization evaluations.

**Table 2.** Overview of simulation cases. The final four cases are additional to determine the optimal phase offset.

| Case | T1 | T2 | T3 |
|---|---|---|---|
| T1 Only (Helix) | Helix | - | - |
| BL Helix | Helix | BL | BL |
| $\varphi_{\text{off}} = 0°$ | Helix | Sync + 0° | BL |
| $\varphi_{\text{off}} = 90°$ | Helix | Sync + 90° | BL |
| $\varphi_{\text{off}} = 180°$ | Helix | Sync + 180° | BL |
| $\varphi_{\text{off}} = 270°$ | Helix | Sync + 270° | BL |
| $\varphi_{\text{off}} = 120°$ | Helix | Sync + 120° | BL |
| $\varphi_{\text{off}} = 150°$ | Helix | Sync + 150° | BL |
| $\varphi_{\text{off}} = 210°$ | Helix | Sync + 210° | BL |
| $\varphi_{\text{off}} = 330°$ | Helix | Sync + 330° | BL |

3. **Synchronization Cases** ($\varphi_{\text{off}} = 0° - 270°$)**:** T2 applies synchronized Helix control with phase shifts of $0°, 90°, 180°$, and $270°$.

4. **Additional Phase Shifts:** Further cases with phase shifts of $120°, 150°, 210°$, and $330°$ refine the optimal phase offset analysis.

Each case has a total simulation time of 2700 seconds, where during the first 600 seconds T2 operates using BL control such that the Helix wake from T1 can propagate through the domain. At $t = 600$ s, the synchronization controller is activated on T2.

Simulations were executed on the Dutch national high-performance computing system Snellius (SURF, 2024), utilizing 512
cores and consuming 24k CPU hours per simulation.

### 3.3.4 Quantities of Interest

The performance metrics for evaluating the synchronization strategies are listed below:

**Estimator Performance Metrics**

– **Root Mean Square Error (RMSE):** Evaluates the deviation between estimated and ground truth phase shifts for tilt
and yaw components, indicating estimation accuracy.

– **Phase Coherence:** Measures the correlation between estimated and true phase shifts at the Helix frequency, providing insight into estimator reliability.

– **Phase Error:** Quantifies the average deviation in degrees between estimated and ground truth phase shifts to assess tracking precision.



**Turbine Performance Metrics**

- **Power Production:** Power output from T2 and T3 to assess the influence of phase synchronization on energy production.

- **Structural Loads:** Damage Equivalent Loads (DELs) on T2 and T3, calculated using rainflow counting, to evaluate effects on fatigue life. For this, we use NREL's Mlife toolbox (Hayman, 2012).

**Flow Analyses**

- **Wake Centerlines:** Extracted using Gaussian convolution on phase-averaged velocity fields to analyze wake deflection and mixing patterns.

- **Velocity Deficits:** Quantified as the reduction in velocity relative to free-stream conditions, providing insights into wake recovery and energy availability.

These metrics collectively offer a comprehensive assessment of the trade-offs and effectiveness of the synchronization control strategies.

## 4 Results

This section presents the results of this study. First, the results of the proposed estimator after validation against the ground truth are presented. Then, we investigate the closed-loop results and the effect on power production and loads for the synchronized test cases. Lastly, flow analysis is performed, investigating wake centerlines and velocity deficits on phase-averaged data.

### 4.1 Estimator validation using ground truth

This section provides a preliminary validation of the estimator, by verifying that it is able to capture the parameters of the incoming Helix wake parameters when T2 is operating with baseline control, i.e. where the closed-loop control on T2 is not applied yet. To do so, we present in Fig. 9 the wind speeds and Helix phases computed by the estimator of the second turbine in the BL Helix case. We compare these estimates with the ground truth as defined in Section 3.3.1.

Regarding the collective wind speed estimates $U_{\mathrm{col}}$, two comments arise. The estimator is able to capture the variations of mean wind speed but does it with a bias. This discrepancy is mostly attributed to differences between the ALM used in the simulations and the BEM used in the estimator (Coquelet et al., 2024b). The ALM tends to compute higher loads than the BEM for similar conditions, especially at relatively coarse grid resolutions as those employed here.

When it comes to the tilt and yaw wind speed components, a transient can be observed for the signals to converge to the ground truth (the estimation process starts at $t = 600\,\mathrm{s}$). From $1000\,\mathrm{s}$, the estimation aligns with the ground truth, even if the amplitudes of the estimated tilt and yaw components tend to be smaller than those of the ground truth. This can be a consequence of the band-pass filtering process applied during the analysis, or of the use of the same amplitude factor $A_{\mathrm{helix}}$ for



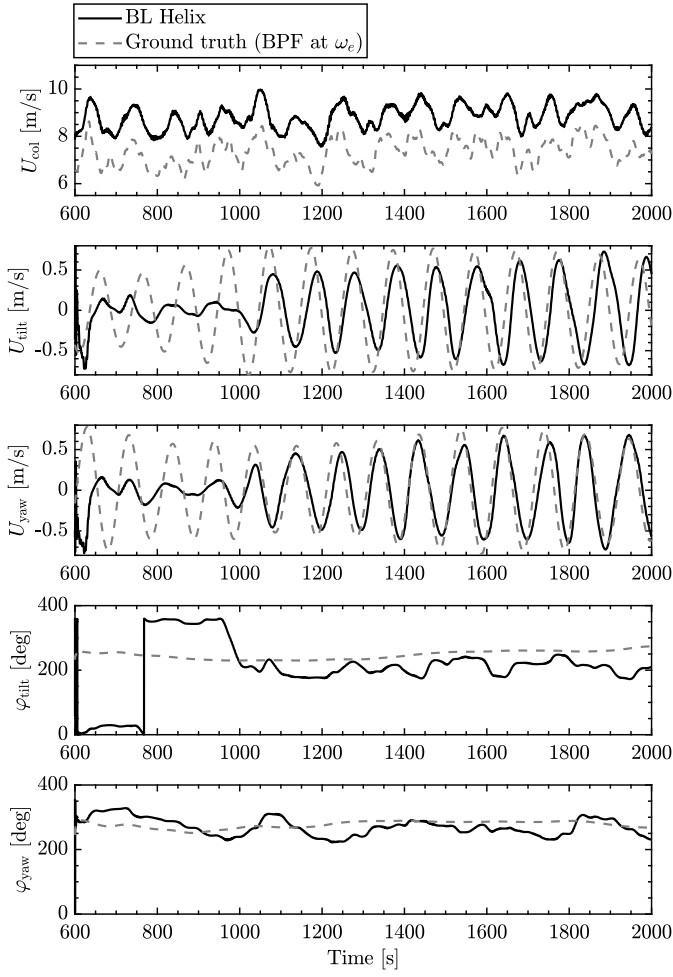

**Figure 9.** Validation of wind speed estimator against ground truth data. Estimation results on T2 for BL Helix case.

tilt and yaw wind speed, limiting flexibility in capturing asymmetries in wake dynamics. Allowing independent scaling factors could improve phase and amplitude accuracy, although, in our case, better results were obtained with a single amplitude factor.

Eventually, the main variables of interest in this case are the estimated phase shifts $\varphi_{\text{tilt}}$ and $\varphi_{\text{yaw}}$. Indeed, these are the values that will be used in the controlled cases. The ground truth values of these signals are obtained from the tilt and yaw wind speed signals using Hilbert transforms. After the estimation transient, the estimates show a consistent trend over time and with the ground truth (quantification is provided in the next section).

These first observations of the estimator's outputs when T2 is operating with baseline control are promising. However, the
question remains: can the estimator maintain this performance when T2 is dynamically pitching? To address this, we integrate the estimator into a closed-loop control framework and quantitatively evaluate whether it consistently captures the phase trends under dynamic pitching conditions.



## 4.2 Estimator Performance Analysis

The goal of this analysis is to quantitatively validate the estimator's performance when integrated within a closed-loop control
system under dynamically varying conditions using the estimator performance metrics defined in Sec. 3.3.4. This section
presents the estimator's performance across synchronization scenarios characterized by distinct phase offsets ($\varphi_{\mathrm{off}}$). The aim
is to confirm that the estimator accurately tracks incoming wake phase shifts ($\varphi_{\mathrm{tilt}}$ and $\varphi_{\mathrm{yaw}}$) and remains effective regardless
of dynamic control actions.

Figure 10 shows the results from the estimator across all four closed-loop synchronized scenarios. A qualitative observation
shows that the estimator closely follows the phase trends, regardless of the downstream control actions. However, some devi-
ations are apparent, particularly for the tilt phase estimate, where more time is required to converge to the correct phase shift
after activating the synchronization controller at $t = 600\,\mathrm{s}$, and larger differences between the estimates can be found compared
to the yaw phase estimate. Despite these anomalies, the estimator performs well overall, maintaining close alignment with the
expected phase shifts according to the ground truth.

To quantify the estimator's reliability, Table 3 summarizes the relevant frequency-domain and phase-domain metrics across
different phase offsets. Phase error indicates the estimator's ability to align with the timing of the ground truth oscillations.
High coherence values (close to 1) suggest a strong correlation between estimated and true signals, reinforcing estimation
reliability. The Root Mean Square Error (RMSE) metric evaluates the estimator's capacity to track the phase over time.

The observed RMSE values for tilt phase estimates, between 30° and 82°, indicate a notable bias in the phase estimation. The
bias in yaw phase estimates is much more constant, between 28° and 45°, indicating stronger estimator consistency. However, it
is important to note that the proposed synchronization control strategy relies predominantly on maintaining a consistent relative
phase offset between turbines, rather than achieving absolute phase accuracy. As the bias remains relatively constant across
scenarios for the yaw phase, its impact on control effectiveness is limited. The consistency allows the controller to reliably
synchronize downstream actions relative to the incoming wake phase, ensuring effective wake mixing. For the tilt phase, less
consistency implies that the synchronization effect may be less predictable between the different cases, although this is not
directly apparent from the results. This is further supported by the observed correlation between applied phase offsets and
power production changes (discussed later on), suggesting that the estimator, despite its bias, still captures the essential phase
dynamics required for successful synchronization.

Further sensitivity studies may be required to analyze the effects of the biases on the controller performance. Nonetheless,
future improvements could focus on systematic bias correction. For instance, calibrating the estimator using controlled field
experiments or enhancing the dynBEM model to better match observed wake dynamics could reduce this bias. Alternatively,
adaptive filtering techniques that account for model-structure uncertainties may further improve phase accuracy.

Overall, these results confirm that the estimator performs robustly across the phase offsets, with coherence levels above 0.8
for most components indicating strong reliability. However, discrepancies in tilt component estimates, particularly in RMSE,
suggest potential improvement areas. These deviations may originate from model inaccuracies within the estimator but could
also be due to uncertainties in the ground truth calculations.



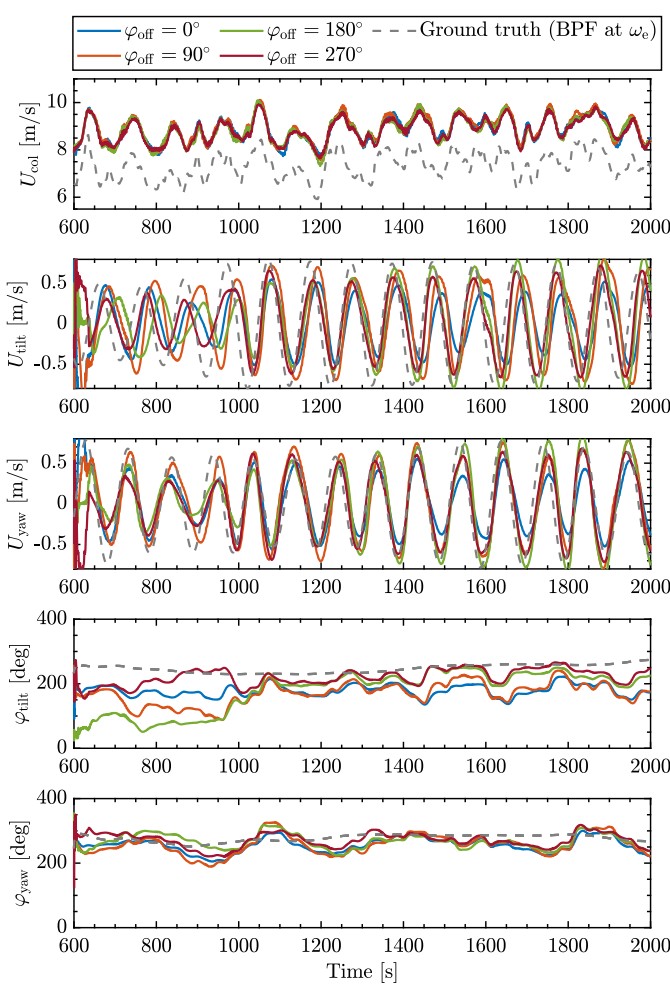

**Figure 10.** Validation of wind speed estimator in closed-loop scenarios. Estimation results on T2 for synchronization cases with different $\varphi_{\mathrm{off}}$.





**Table 3.** Frequency-domain and EKF phase metrics for different orientations at $f_0$.

| Metric | 0° | 90° | 180° | 270° |
|---|---|---|---|---|
| *Frequency-Domain Metrics* | | | | |
| $U_{\mathrm{col}}$ Phase Error [deg] | 2.95 | 3.35 | 7.75 | 4.98 |
| $U_{\mathrm{col}}$ Coherence | 0.87 | 0.82 | 0.83 | 0.89 |
| $U_{\mathrm{tilt}}$ Phase Error [deg] | -16.85 | -6.58 | 16.83 | 5.03 |
| $U_{\mathrm{tilt}}$ Coherence | 0.65 | 0.77 | 0.94 | 0.96 |
| $U_{\mathrm{yaw}}$ Phase Error [deg] | 0.07 | 12.36 | 13.50 | 8.00 |
| $U_{\mathrm{yaw}}$ Coherence | 0.88 | 0.88 | 0.94 | 0.91 |
| *Phase Metrics (EKF Estimates)* | | | | |
| $\varphi_{\mathrm{tilt}}$ RMSE [deg] | 82.37 | 68.99 | 39.33 | 30.72 |
| $\varphi_{\mathrm{yaw}}$ RMSE [deg] | 44.78 | 46.11 | 34.33 | 27.75 |

## 4.3 Closed-loop synchronization analysis

To understand the effect of synchronization on the turbine operation, we examine the correlations between the estimated incoming wake, the pitch action that is performed, and the impact of the loads. Figure 11 presents these signals for the tilt axis. It first shows the control signals generated for the downstream turbine based on the estimated phase and the applied additional phase shift, following Eq. 22. The control signals are evenly spaced in accordance with the intended phase shifts, indicating that the control system effectively implements the intended phase adjustments. This regular spacing also confirms that the estimator's phase outputs are being correctly interpreted by the control system and that the phase shifts are accurately applied to the downstream control actions. Figure 11 then shows the impact of the pitch action on the moments. Note that the moments have been low-pass filtered with a passband frequency 0.01 Hz for clarity. In the BL Helix case, T2 does not perform the Helix control and its pitch angles remain constant during operation. The moments are therefore the direct reflection of the local changes in wind speed. When the incoming $U_{\mathrm{tilt}}$ shows a local maximum (as at $t = 1275$ s for example), the tilt moment $M_{\mathrm{tilt}}$ also reaches a local maximum. When synchronization is applied, the effect of the pitch interacts with that of the local changes in wind speed. We here highlight two characteristic interactions:

- $\varphi_{\mathrm{off}} = 180°$: The pitch acts in opposition of phase with the incoming wind, as described in Section 2.6. When the wind speed perturbation $U_{\mathrm{tilt}}$ reaches a maximum, the pitch angle $\beta_{\mathrm{tilt}}$ reaches a minimum. From the aerodynamic perspective, this means that the angles of attack increase, and so do the moments. This is confirmed when looking at the signal of the tilt moment $M_{\mathrm{tilt}}$, whose peak appears at the same location as in the Helix BL case (black curve), but with a higher amplitude. This amplification reflects a constructive interference between the control action and the oscillations in incoming wind speeds induced by the Helix wake.



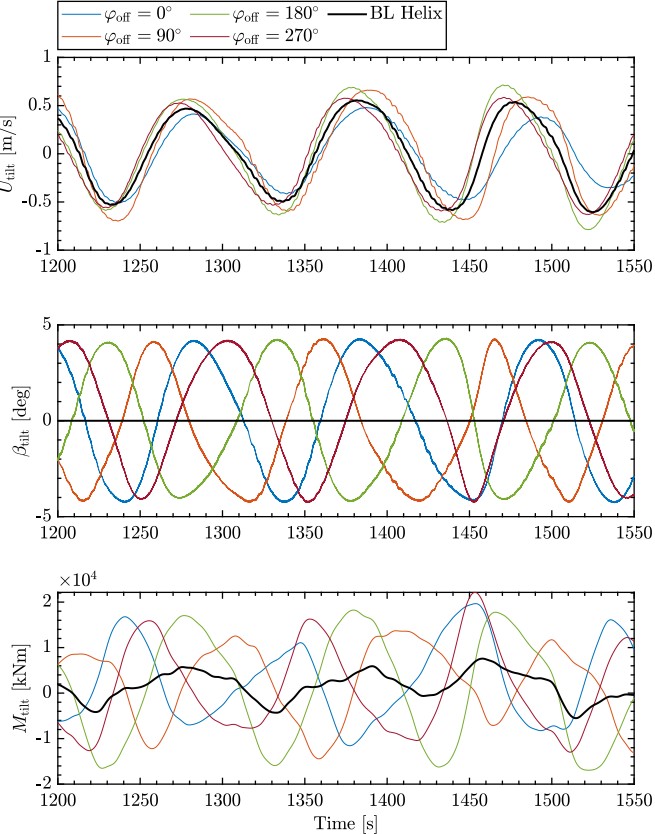

**Figure 11.** Tilt wind speed estimate $U_{\text{tilt}}$, synchronized pitch $\beta_{\text{tilt}}$ and resulting moment $M_{\text{tilt}}$ of the downstream turbine under various phase-shifted control strategies.

- $\varphi_{\text{off}} = 0°$: The pitch acts in phase with the incoming wind. When the wind speed perturbation $U_{\text{tilt}}$ reaches a maximum, the pitch angle $\beta_{\text{tilt}}$ also reaches a maximum. This implies a decrease in local angles of attack, and hence of moments. There is, therefore, a competing effect: the blade passes through a high wind speed region but reduces its angle of attack. In this case, the pitch angle is rather important ($4°$) and the control effect takes over the incoming wind effect. Looking at the time series of the moment, one can observe that the tilt moment signal is in this case in opposition of phase with that of the BL Helix case (black curve). There is therefore a destructive interference between the control action and the incoming Helix wake.

Figure 12 examines the frequency content of the tilt moments displayed in Fig. 11, focusing on the frequency band around the helix frequency peak. The 180° phase shift case shows the largest magnitude at the helix frequency. This result indicates a stronger resonant effect when the control signals are phase-shifted by 180°, which could be linked to the synchronization of the downstream turbine's response with the periodic wake structure. Such resonance may lead to both beneficial and adverse




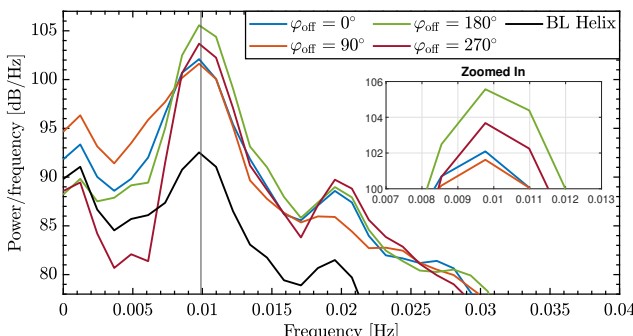

**Figure 12.** Frequency content analysis of tilt moments around the Helix frequency. The 180° case exhibits higher magnitudes at the Helix frequency, indicating enhanced resonance.

effects, enhancing power production but potentially increasing fatigue loads. The next section will further investigate the performance differences between the different synchronization cases from a power and load perspective.

### 4.4 Closed-loop performance analysis

The effect of these phase-shifted control strategies on overall power production is shown in Fig. 13. In addition to the relative power increases plotted by black crosses, a Gaussian Process (GP) regression fit is plotted that captures the trend between these data points and provides confidence intervals (see e.g. Rasmussen (2004)). This approach not only interpolates between the measured phase shifts but also quantifies the predictive uncertainty, illustrating the potential power gains at unmeasured offsets. It appears that, from this figure, the optimal offset is at 150°, which yields a large power gain of around 10% on the third

turbine. Collectively, this amounts to a power increase of around 5%, since a small loss on T2 can be observed. Interestingly, the best-performing case also sees the lowest power loss on T2, while the worst-performing cases (270° and 330°) exhibit significant power losses for both T2 and T3—over 6% overall—suggesting that implementing a Helix without synchronization on T2 could lead to considerable power losses. Overall, the results highlight that the synchronized Helix wake mixing approach exhibits both optimal and suboptimal regions of power production governed by the phase shift relative to the incoming Helix

wake.

Figure 14 provides an overview of the load impacts across different turbine components due to the various phase-shifted control strategies. The results reveal that the loads on the second turbine generally increase for all phase shifts, except a notable tower base load reduction in both fore-aft and side-side is observed for the best-performing 150° case. For the third turbine, increased loads are primarily observed in cases with power gains. This is driven by enhanced wake mixing, which amplifies

wake meandering at T2, causing greater wind speed fluctuations at T3, this is further highlighted in Section 4.5. Such variations increase cyclic loading and fatigue, leading to higher DELs. These results stress the trade-off between power production and structural loading, where certain phase shifts that improve power output might simultaneously increase the fatigue build-up on the turbines, potentially affecting their lifespan.



**Figure 14.** Effect of phase offsets in the synchronization controller on damage-equivalent loads for both T2 and T3. The data points show the relative DEL change, while the Gaussian process confidence interval shows the uncertainty of the DEL change at untested phase offsets. CI denotes confidence interval.

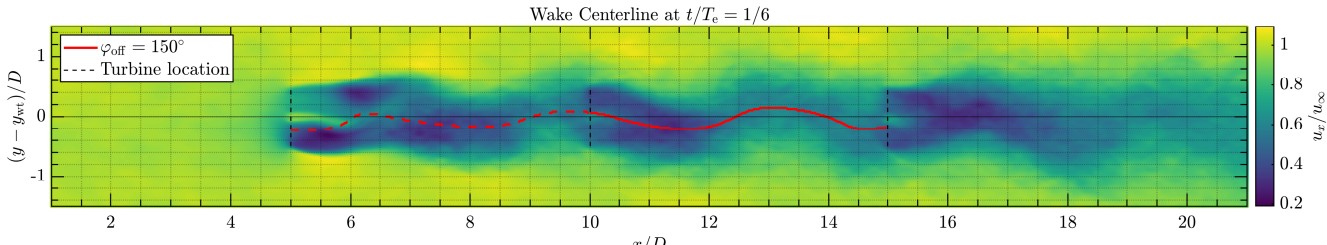

(a) 150° phase shift case. The wake centerline (red line) exhibits sustained periodicity and constructive interference, improving wake mixing and downstream power production.

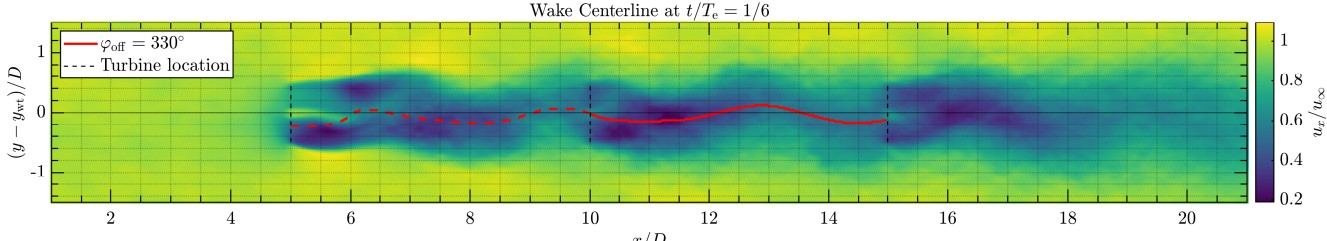

(b) 330° phase shift case. The wake trajectory (red line) displays flattened oscillations and destructive interference, reducing wake mixing and downstream performance.

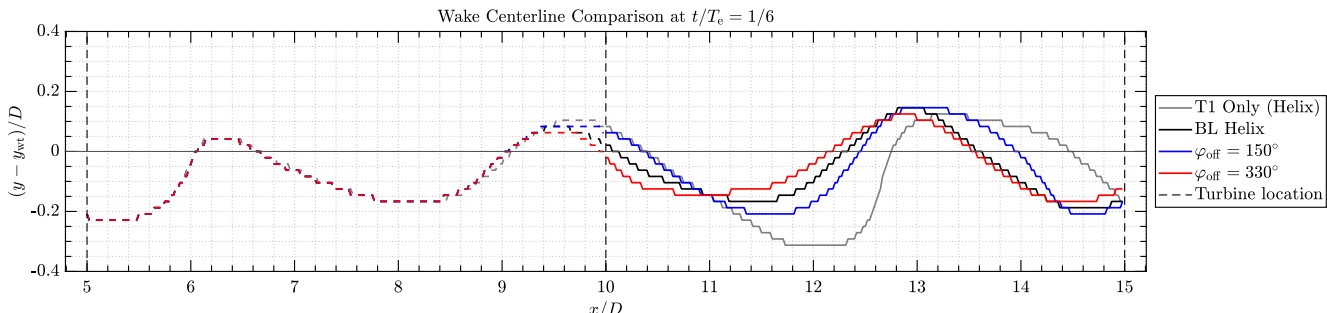

(c) Wake trajectory comparison between the best-performing (150°) and worst-performing (330°) cases, alongside the baseline Helix wake trajectory.

**Figure 15.** Comparison of phase-averaged horizontal velocity fields at hub height for different phase shift cases. $T_e$ is the Helix period and $t$ indicates the time instance at which the snapshot is taken. The best-performing case (150°) aligns closely with the natural Helix oscillation with T1 only, while the worst-performing case (330°) diverges. Note that the $y$-axis has been scaled in the third subfigure to better visualize the differences.





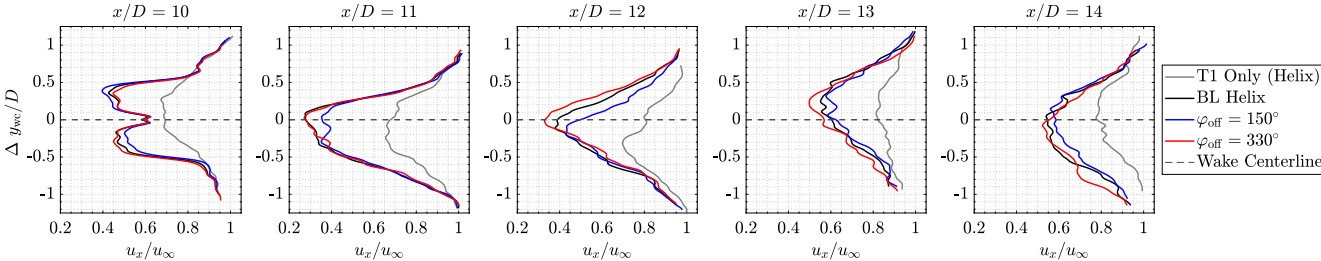

**Figure 16.** Velocity deficits across several flow slices downstream of T2 for the best (150°) and worst (330°) phase-shift cases. The best-performing case shows reduced deficits near the wake centerline, suggesting improved wake recovery.

Lastly, in Fig. 16, we examine the velocity deficits across the wake's width at several instances behind T2. To obtain these results, the wake centers have been recentered to zero using the wake centerline. This enables us to separate the effects of wake displacement and wake deficit reduction when it comes to the increased power on T3. This figure shows that the in-phase control ($\varphi_{\mathrm{off}} = 150°$) not only enhances the lateral/vertical displacement of the downstream wake but also reduces the intensity of the wake deficit. This 150° case enhances the combined effects of the helix: it promotes wake deficit recovery through mixing, but also enhances lateral wake displacement, which could be assimilated to forced/enhanced meandering. These combined effects explain the higher power leveraged at T3.

## 5 Discussion

This study presents a significant advancement in phase synchronization strategies for wind farm optimization, addressing research gaps identified in prior research by Korb et al. (2023) and van Vondelen et al. (2024b).

- **Dynamic Phase Synchronization:**

  - Korb et al. (2023) demonstrated that specific phase alignments between turbines could improve power recovery. However, their approach, which relied on geometric phase shifts, is sensitive to wind speed and turbine spacing, limiting practical applicability.

  - Our EKF-based synchronization method addresses this gap by dynamically estimating the upstream wake phase and adjusting downstream turbine controls accordingly. The approach is robust to changing wind conditions and turbine configurations.

  - The results indicate that a 150° phase shift between the incoming wake and the turbine action achieved a substantial 10% increase in power output at the third turbine (T3), with an additional overall power gain of 5% over the baseline achieved by implementing the Helix.

- **Balancing Power Gains and Structural Loads:**




- While power gains were accompanied by increased structural loads, particularly on T2, this trade-off provides valuable insights for optimizing load-mitigation control strategies in future applications.

- Understanding the relationship between phase alignment, power gains, and structural loads enables more informed control decisions, ensuring long-term turbine integrity while maximizing energy production.

  - **Enhanced Performance Over Previous Studies:**

    - Compared to the +6% power gain reported by van Vondelen et al. (2024b) using in-phase synchronization, our method (+10% on T3) demonstrates superior performance by leveraging optimized out-of-phase alignments.

    - Although this approach introduces additional complexity in phase estimation and control, it offers greater potential for enhancing wind farm efficiency.

  - **Insights from Flow Dynamics:**

    - Flow analysis reveals that constructive interference (e.g., 150° phase offset) sustains the natural Helix wake oscillation, enhancing wake recovery and improving downstream power generation.

    - Conversely, destructive interference (e.g., 330° offset) disrupts wake recovery, reducing downstream energy yield. This highlights the importance of aligning control strategies with the natural dynamics of the wake.

The main challenge for practical implementation lies in accurately parameterizing wake dynamics and ensuring robust phase detection under high turbulence and gust conditions. While the proposed EKF-based method demonstrates reliable performance in controlled scenarios, increased turbulence could make identifying consistent phase trends more difficult, complicating syn-
chronization. Enhancing model fidelity and integrating adaptive estimation techniques may be necessary to ensure robustness in variable conditions. Wind tunnel and field testing will be critical to confirm real-world applicability.

Overall, the proposed EKF-based synchronization method demonstrates promise for real-world applications, offering a robust and adaptable approach to enhance wind farm performance. Future research could focus on further refining load-mitigation control strategies and expanding the methodology to larger wind farm configurations.

## 6 Conclusion

This study proposed and evaluated an Extended Kalman Filter-based phase synchronization method to enhance downstream turbine performance in wind farms through coordinated wake control. By addressing the limitations of linear Kalman filters and incorporating a dynamic Blade Element Momentum model, the approach demonstrated improved accuracy in estimating wake phases and collective wind speeds, which are necessary for synchronized control strategies.
The results of this study performed in the CNBL with $U_{\mathrm{hub}} = 10.5$ m/s and $TI_{\mathrm{hub}} = 3\%$ showed that the optimal phase shift yields a significant net power gain of approximately 10% at turbine 3 and 5% collectively across turbine 2 and turbine 3 while sustaining the natural Helix oscillation to enhance wake recovery. The findings also reveal a trade-off between power





gains and increased structural loads, particularly on turbine 2. This suggests the need for a careful balance between energy production and turbine fatigue build-up in wind farm control.

Future efforts could enhance the estimator to address model discrepancies and robustness, explore adaptive approaches to mitigate structural loads while maintaining high power yields and expand on testing scenarios including different wind farm layouts.

*Code availability.* Code and data are available upon request.

*Author contributions.* AAWvV led the conceptualization, methodology, software development, validation, formal analysis, investigation,
visualization, and original draft preparation. MC implemented the dynBEM model in the estimator and contributed to conceptualization, methodology, validation, visualization, manuscript review, and editing. STN contributed to methodology development and supervision and participated in manuscript review and editing. JWvW contributed to conceptualization, supervision, funding acquisition, and manuscript review and editing. All authors provided feedback on the methodology and reviewed the final manuscript.

*Competing interests.* At least one of the (co-)authors is an Associate Editor of Wind Energy Science.

*Acknowledgements.* This work is part of the Hollandse Kust Noord wind farm innovation program where CrossWind C.V., Shell, Eneco, Grow, and Siemens Gamesa are teaming up; funding for the PhDs was provided by CrossWind C.V. and Siemens Gamesa. We further would like to acknowledge the computing resources provided by DelftBlue and thank SURF for the computational time made available on the Dutch national supercomputer Snellius (grant number: EINF-10024).



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
