# Peer review of "Synchronized Helix Wake Mixing Control"

_Wind Energy Science, 2025_

## Referee Comment (RC1)

The reviewer strongly believes that the paper presents insights into synchronized Helix wake mixing control. The results appear original and well written.

Page 1, Lines 15–20 (Introduction): When citing Manwell et al. (2010) and Barthelmie et al. (2009), clarify whether the 20% drop applies to onshore or offshore farms—or both—as atmospheric stability differs markedly between environments.

Page 2, Lines 30–35 (Introduction): The comparison between DIC and Helix (lower tower loads vs. higher gains) lacks quantitative value.

Page 2, Lines 46: The phrase "deeper arrays" is used without definition. Indicate the number of turbine rows or array dimensions to which "deeper" refers (e.g., > 5 rows).

Page 4, Figure 1 Caption: The caption omits key LES conditions (Reynolds number, TSR, inflow laminar vs. turbulent). Please add "TSR, Re number" to the caption for reproducibility.

Page 6, Lines 124: You state that $\omega_r \pm \omega_e$ yields the effective rotating-frame frequency. A brief note on potential aliasing when $\omega_e$ approaches $\omega_r$ (say when $\omega_e \approx 0.9\,\omega_r$) would alert practitioners to choose safe Strouhal ranges as explained in Equation 2.

Page 7, Equation (11): Please confirm if random wall model for uuK as described is applicable for oscillatory models.

Page 14, Algorithm 1, Step 1: which states that "Identify the frequency band of interest". It will be helpful to state or refresh the reader, or which criteria are being considered.

Page 14, Lines 305: In Equation (22), amplitude A is reused from upstream, but pitch rate constraints can vary downstream. Please comment on how actuator saturation is handled.

Section 4.4 – Figure 13, The Gaussian Process fit effectively interpolates power gains, but the manuscript does not specify the kernel choice, hyperparameter tuning method, or the confidence-interval level (e.g., 95%). Including these details (perhaps in a brief footnote) would allow other researchers to reproduce the interpolation for better judgment.

General comments

Authors should consider scaling down the scope of the manuscript or splitting into two papers, as it becomes hard to follow at some point. The manuscript tackles estimation theory, control design, high-fidelity LES validation, fatigue assessment, and flow-recovery analysis all in one paper, which makes it difficult to follow the core contributions.

---

## Author Comment (AC1)

Date June 11, 2025 Our reference n/a Your reference n/a Contact person A.A.W. van Vondelen Telephone/fax +31 (0)15 27 86707 / n/a E-mail A.A.W.vanVondelen@tudelft.nl Subject Response to Referees

**Delft University of Technology**

Delft Center for Systems and Control

Address Mekelweg 2 (3ME building) 2628 CD Delft The Netherlands

www.dcsc.tudelft.nl

Anonymous Referee #1 Anonymous Referee #2 *Referees, Wind Energy Science*

Dear Referees,

We sincerely thank you for your valuable and positive feedback on our paper. Your comments have greatly assisted us in enhancing the quality of our work. We have carefully considered all of the points raised and have made adjustments to the paper. This letter serves to address your feedback and provide an overview of the changes made. Below, we will respond to each of your review reports.

Yours sincerely,

Aemilius van Vondelen Marion Coquelet Sachin Navalkar Jan-Willem van Wingerden

Enclosure(s): Response to comments of Anonymous Referee #1 Response to comments of Anonymous Referee #2

Page/of 1/15

**Response to comments of Anonymous Referee #1**

- Referee: The reviewer strongly believes that the paper presents insights into synchronized Helix wake mixing control. The results appear original and well written.
- Authors: We thank the Referee for the kind words and for taking the time to review our manuscript.
- Referee: Page 1, Lines 15–20 (Introduction): When citing Manwell et al. (2010) and Barthelmie et al. (2009), clarify whether the 20% drop applies to onshore or offshore farms—or both—as atmospheric stability differs markedly between environments.
- Authors: We thank the Referee for this suggestion. In the revised manuscript, we now clarify that the 20% power drop cited from [2] specifically applies to large offshore wind farms.
- Referee: Page 2, Lines 30–35 (Introduction): The comparison between DIC and Helix (lower tower loads vs. higher gains) lacks quantitative value.
- Authors: We thank the Referee for pointing out the need for quantitative comparisons. In the revised manuscript, we now include performance and fatigue load metrics from [3] to compare between Dynamic Induction Control (DIC) and the Helix strategy.
- Text excerpt: This approach has demonstrated moderate power gains (up to 4.6% on T2 in two-turbine setups with 2.5 degree pitch amplitude) [3], albeit with substantially increased tower load variations, up to 104% higher tower DELs compared to base-line operation under low turbulence intensity conditions [3].

A later method involves rotating the thrust vector across the rotor disk, creating a helical wake shape [4]. Compared to DIC, the Helix approach results in lower tower load variations and higher power gains, garnering considerable attention [3, 10]. For instance, the counterclockwise Helix implementation achieved a 12.1% increase on T2 with 2.5 degree pitch amplitude while increasing tower loads by only 11% under similar conditions. On downstream turbines (which are impinged by a Helix or DIC wake while operating at baseline control), Helix was also shown to induce 5–10% lower fatigue loads than DIC. One major challenge remains the pitching frequency of the actuators. While similar to that of IPC, DIC has significantly lower pitch variations, limiting damage to the pitch bearings significantly. Nevertheless, these findings demonstrate a more favorable trade-off between performance and structural loading for Helix wake mixing.

Referee: Page 2, Lines 46: The phrase "deeper arrays" is used without definition. Indicate the number of turbine rows or array dimensions to which "deeper" refers (e.g., 5 rows).

Date June 11, 2025 Our reference n/a Page/of 3/15

- Authors: We thank the Referee for pointing this out. We have added context on wake recovery in deeper arrays from large-eddy simulations [8], which motivates a focus on the first two to three rows for active wake mixing strategies.
- Text excerpt: While [5] and [9] have shown that phase differences between periodic wakes in a multi-turbine setup can influence power production, a robust synchronization method remains lacking. Moreover, even without active control, wake deficits naturally recover through entrainment and wake-to-wake interactions: large-eddy simulations indicate that by the fourth to sixth turbine row, the average velocity deficit has already recovered substantially [8]. Consequently, dynamic wake-mixing strategies yield their greatest benefit in the first two to three rows, where the deficit is strongest. Extending phase-synchronized Helix control to these upstream rows is therefore essential to maximize farm-level performance. Hence, it is important to address this research gap.
  - Referee: Page 4, Figure 1 Caption: The caption omits key LES conditions (Reynolds number, TSR, inflow laminar vs. turbulent). Please add "TSR, Re number" to the caption for reproducibility.
  - Authors: We appreciate the Referee's attention for reproducibility. Figure 1 is adapted from [6], which based it on data from [4]. As the specific values for Reynolds number, tipspeed ratio, and inflow conditions are not explicitly stated in the original publication [6] and the figure is not present in [4], we have updated the caption to refer to these papers directly for further details. Note that it was already stated that the simulation has laminar inflow.
  - Referee: Page 6, Lines 124: You state that  $\omega_r \pm \omega_e$  yields the effective rotating-frame frequency. A brief note on potential aliasing when  $\omega_e$  approaches  $\omega_r$  (say when  $\omega_e \approx 0.9\omega_r$ ) would alert practitioners to choose safe Strouhal ranges as explained in Equation 2.
  - Authors: We thank the Referee for this observation. In practice, however, the recommended Strouhal range prevents  $\omega_e$  from approaching  $\omega_r$ . Since

$$\frac{\omega_e}{\omega_r} = \frac{2\pi f_e}{\omega_r} = \frac{\pi \operatorname{St} U_{\infty}/D}{\operatorname{TSR} (U_{\infty}/D)} = \frac{\pi \operatorname{St}}{\operatorname{TSR}},$$

even a conservative choice of  $\text{St} \leq 0.4$  with a typical turbine  $\text{TSR} \approx 6$  yields  $\omega_e/\omega_r \leq 0.21$ , i.e.  $\omega_e \ll \omega_r$ . This ensures that neither the difference  $\omega_r - \omega_e$  nor the sum  $\omega_r + \omega_e$  reaches zero frequency or the actuator's low-frequency cutoff. We have added a remark in the text to make this clear.

- Text excerpt: Note that  $\omega_e/\omega_r = \text{St}/\text{TSR}$ , choosing  $\text{St} \le 0.4$  for a typical tip speed ratio (TSR)  $\text{TSR} \approx 6$  ensures  $\omega_e/\omega_r \le 0.21$  and thus stays far from any low-frequency aliasing.
  - Referee: Page 7, Equation (11): Please confirm if random wall model for uuK as described is applicable for oscillatory models.

Date June 11, 2025 Our reference n/a Page/of 4/15

Authors: We thank the Referee for this observation. In our EKF formulation, the state vector  $\mathbf{u}^{\mathrm{u}} = [A_{\mathrm{col}}, A_{\mathrm{helix}}, \varphi_{\mathrm{tilt}}, \varphi_{\mathrm{yaw}}]^{\mathsf{T}}$  is indeed assumed to follow a random walk. The periodic behavior, the sine-wave at frequency  $\omega_e$  with amplitude  $A_{\mathrm{helix}}$  and phase shifts  $\varphi$ , enters only through the nonlinear measurement function  $h(\mathbf{u}^{\mathrm{u}}, \mathbf{u}^{\mathrm{c}})$ , which includes

$$U_{\text{tilt}} = A_{\text{helix}} \sin(\omega_e t + \varphi_{\text{tilt}}), \quad U_{\text{yaw}} = A_{\text{helix}} \cos(\omega_e t + \varphi_{\text{yaw}})$$

and then maps these into blade-load predictions via dynBEM and the MBC transform. Thus, the random walk on  $\mathbf{u}^{u}$  only allows slow drift of the underlying sinusoid's amplitude and phase, capturing changes in inflow, while the measurement function itself generates the full oscillatory load signal. This avoids the need to augment the state with explicit oscillator dynamics and yields direct, online estimates of the Helix phase for real-time synchronization. We have added a clarifying sentence to explain that  $h(\cdot)$  includes both the sine-wave windspeed model and the subsequent dynBEM model.

Text excerpt: In practice,  $h(\cdot)$  first uses the estimated amplitude and phase to generate the fixed-frame periodic wind perturbations,

$$U_{\text{tilt}} = A_{\text{helix}} \sin(\omega_{\text{e}} t + \varphi_{\text{tilt}}), \quad U_{\text{yaw}} = A_{\text{helix}} \cos(\omega_{\text{e}} t + \varphi_{\text{yaw}}),$$

and then maps these through the backward MBC transform and dynBEM to predict blade loads, thereby embedding both the oscillatory wake model and the turbine dynamics into the measurement.

- Referee: Page 14, Algorithm 1, Step 1: which states that "Identify the frequency band of interest". It will be helpful to state or refresh the reader, or which criteria are being considered.
- Authors: We thank the Referee for this suggestion. In the revised manuscript, Algorithm 1, Step 1 now explicitly states that the band of interest is around the Helix excitation frequency  $\omega_e$ , ensuring that the subsequent noise extraction focuses on frequencies outside the Helix bandwidth.
- Text excerpt: Identify the frequency band of interest in the signal (around  $\omega_e$  in our case).
  - Apply a high-pass filter to isolate high-frequency noise (see Fig. 5).
  - Referee: Page 14, Lines 305: In Equation (22), amplitude A is reused from upstream, but pitch rate constraints can vary downstream. Please comment on how actuator saturation is handled.
  - Authors: We thank the Referee for this point. We have updated Equation (22) to use the notation  $A_{T2}$  for the downstream turbine's Helix amplitude, explicitly noting that it can differ from the upstream value to respect local actuator limits. We also added a sentence clarifying that all pitch commands, including those using  $A_{T2}$ , are passed through ROSCO's [1] pitch-rate limiter to automatically clip any command exceeding the downstream turbine's rate or angle constraints, thereby preventing actuator saturation.

where  $A_{\rm T2}$  is the downstream turbine's amplitude,  $\omega_{\rm e}$  is the excitation frequency, and  $\varphi_{\rm off}$  is an additional phase shift that can be applied to modify the alignment of the up-and downstream Helix wakes. Note that the amplitude can differ from the upstream amplitude, and pitch commands are fed through ROSCO's pitch rate limiter to prevent actuator saturation.

- Referee: Section 4.4 Figure 13, The Gaussian Process fit effectively interpolates power gains, but the manuscript does not specify the kernel choice, hyperparameter tuning method, or the confidence-interval level (e.g., 95%). Including these details (perhaps in a brief footnote) would allow other researchers to reproduce the interpolation for better judgment.
- Authors: We thank the Referee for this suggestion. The Gaussian Process in Figure 14 (in revised manuscript) employs a squared-exponential (SE) covariance function ('covSEiso' in GPML), with hyperparameters (length-scale and signal variance) optimized by maximizing the log-marginal likelihood via gradient-based training. The shaded region in the plot denotes the 95% predictive confidence interval. A brief footnote to Figure 14 (in revised manuscript) now states these choices clearly.
- Text excerpt: GP configuration: zero-mean prior; squared-exponential covariance (GPML's covSEiso [7]); hyperparameters (length-scale and signal and noise variances) optimized via marginal likelihood maximization (gradient descent); shaded band denotes the 95 % predictive confidence interval.
  - Referee: Authors should consider scaling down the scope of the manuscript or splitting into two papers, as it becomes hard to follow at some point. The manuscript tackles estimation theory, control design, high-fidelity LES validation, fatigue assessment, and flow-recovery analysis all in one paper, which makes it difficult to follow the core contributions.
  - Authors: We understand the concern about the broad scope of the manuscript. Our intention is to address one central question: can downstream turbines synchronize with the upstream wake using only local measurements, and does this improve performance under realistic conditions? To answer this, we bring together different components: estimation, control, high-fidelity simulation, and flow analysis, which we see as closely connected parts of a single framework.

To make the structure easier to follow, we have revised the end of the introduction to clearly explain how each section contributes to the overall goal, and we now include a more detailed outline of the paper to guide the reader (see end of Section 1). Date June 11, 2025 Our reference n/a Page/of 6/15

Text excerpt: This work covers a broad range of topics: estimation, control design, high-fidelity simulation, and flow analysis. These components are integrated to address a central objective: enabling synchronized wake mixing control in a realistic wind farm environment using only local turbine measurements. Each component contributes to this goal. The estimator infers the phase of upstream Helix wake motion; the controller synchronizes the actuation of downstream turbines; high-fidelity simulations provide a realistic testing environment; and fatigue and flow analyses assess the resulting impact on turbine performance and wake development. Together, these elements provide a comprehensive evaluation of the proposed approach. To help navigate the scope of this work, an outline is provided below.

**Section** ?? introduces the estimation and control framework. It describes the baseline Helix control approach, presents the Extended Kalman Filter (EKF) for estimating the upstream wake phase from turbine blade loads, outlines the wake parametrization and the dynBEM-based internal model, explains the noise tuning strategy, and introduces the synchronization controller design.

**Section 3** describes the high-fidelity simulation setup, including the inflow conditions, turbine model, and control implementation. The section also defines the evaluation cases and performance metrics.

**Section 4** presents the results. First, the EKF estimator is validated against the ground truth. Then, the closed-loop control performance is analyzed in terms of power production and structural loading. Finally, flow visualizations and velocity deficit analyses are used to interpret the underlying physical mechanisms of synchronization.

**Section 5** compares the results of this work with earlier studies and discusses the limitations compared to our approach.

Section 6 concludes this work and presents possible future research directions.

**Response to comments of Anonymous Referee #2**

- Referee: "The paper is well written, the subject is interesting and the results appear rather robust. However, some important details about the flow setting, the choice of some relevant parameters and some important measures are missing or unclear, hindering the reliability and interpretation of the results. Please consider the following detailed comments."
- Authors: We thank the Referee for their positive comments and appreciate their time and efforts to review our work.
- Referee: Section 2.2: "Concerning the approximation of the Jacobian by central differences, it is stated that the choice of dn requires balancing truncation and round-off errors. I imagine that this balancing has been done in the calibration of the algorithm, and it is definitely worth showing the results of this calibration in the paper, or justifying in more detail the choice of dn (which is indeed not specified)."
- Authors: We thank the Referee for this helpful comment. In the revised manuscript, we now specify that the central difference step size was set to  $dn = 1 \times 10^{-5}$  rad. This value was chosen empirically: several candidate step sizes were tested, and this one provided the most stable and accurate results in our implementation. A clarifying paragraph was added to the text.
- Text excerpt: The chosen value in our setup is  $dn = 1 \times 10^{-5}$  rad. This value was selected empirically based on implementation testing. It offered the most stable and accurate performance among the values tried. Given that the control input amplitude is approximately 0.07 rad (4 degrees), this perturbation corresponds to about 0.014% of the signal magnitude, small enough to remain in the linear regime while avoiding round-off errors.
  - Referee: Section 2.5: "Figure 5: the cut-off frequency appears too low, since it completely cuts off the strong peak at frequency 0.3, which is definitely not noise. I strongly suggest increasing it to 0.4 at least, and comparing the results."
  - Authors: We understand the Referee's concern and agree with the reviewer that the peak at 3 Hz is not noise. We have clarified this point in the revised manuscript. The low-pass filter applied is intentionally designed to isolate the Helix control frequency, typically around 0.15 Hz. The 0.3 Hz peak corresponds to higher-order dynamic content not used in this control strategy and is considered part of the unmodelled dynamics. Including this component would introduce bias in the estimation. This design choice ensures the estimator operates within the intended control bandwidth.
- Text excerpt: The filter was configured to isolate the Helix control frequency, while suppressing higher-frequency dynamics such as the 0.3 Hz component, which are not used in the estimator and may introduce bias. We employ a high-pass filter on the signal to isolate these parts, extracting the high-frequency components.

Date June 11, 2025 Our reference n/a Page/of 8/15

- Referee: Section 3.1: "Is it not clear whether the tower is taken into account, and how. Also, the domain and boundary conditions are defined only for the precursor simulation. It is not clear why the grid is refined upstream of the first turbine (usually it is refined in correspondence with the turbines). The sampling time of the y-z planes for the inflow boundary is not specified. The Coriolis frequency is also not provided, nor the tip speed ratio. All in all, the computational setting is obscure, and needs more details, as well as validation of the results."
- Authors: We thank the referee for the detailed feedback and have clarified the computational setup. The actuator line method now explicitly includes 72 points for the tower, and it is stated that the turbine simulations use the same domain and boundary conditions as the precursor. The mesh refinement region indeed starts 4.5D upstream of the first turbine to ensure that inflow turbulence and shear are well-resolved before reaching the rotor. The inflow planes at the domain inlet are now specified to be sampled at 1 Hz for 45 minutes. Additionally, we have added the Coriolis frequency corresponding to the simulated latitude and noted the operating tip-speed ratio of the turbines.
- Text excerpt: After this initial phase, y-z planes at the inflow boundary (x = 0 m) are sampled for 45 minutes at a frequency of 1 Hz to serve as inflow conditions for turbine simulations. For these simulations, the domain and boundary conditions remain identical to those used in the precursor run. The turbine blades are modeled using the Actuator Line Method (ALM) coupled with OpenFAST.

The ALM setup includes 60 actuator points per blade and 72 points for the tower. Turbines are operating around a tip-speed ratio of 9.3. The OpenFAST simulations are restarted from a converged precursor checkpoint and advanced synchronously with the LES, using a fixed time step of 0.05s. Inflow planes extracted at 1Hz ensure that the dominant Helix excitation is well-resolved.

The turbines, which are modeled by OpenFAST (see next section), are placed within the domain at coordinates  $(x = 1200 \ m, \ y = 1600 \ m)$  for turbine 1 (T1),  $(x = 2400 \ m, \ y = 1600 \ m)$  for turbine 2 (T2), and  $(x = 3600 \ m, \ y = 1600 \ m)$  for turbine 3 (T3). This corresponds to a 5D spacing, where D represents the rotor diameter (see Table 1), from the inflow and between the turbines, and also sufficient space for wake development behind the third turbine.

To facilitate higher-resolution flow analysis around the wind turbines, a mesh refinement to 5 m is implemented. This refinement covers a static box area starting 4.5D upstream of the first turbine, with dimensions of 5040 m in the *x*-direction, 960 m in the *y*-direction, and 600 m in the *z*-direction. The upstream extension ensures accurate resolution of incoming turbulence and shear from the inflow boundary, providing well-resolved conditions at the rotor plane for synchronization analysis.

Coriolis effects are included via the CNBL setup, using a latitude of  $52.6^\circ$ , corresponding to a Coriolis frequency of approximately  $1.3\times10^{-4}~s^{-1}$ .

- Referee: Section 3.3.1: "The procedure for the wind velocity data is unclear. It is claimed that the wind speed along the sampling lines are averaged over time, but averaging over time along fixed lines would smooth out the dynamics. I guess it is a phase average instead of a time average? Please discuss this point in detail."
- Authors: We thank the reviewer for highlighting that the procedure for ground truth calculation requires clarification. In the text we have added a pseudo-code, showing the full producure of the computation.
- Text excerpt: See Alg 1 on the next page.
  - Referee: Section 4.1: "Figure 9: Why the phase shifts are very smooth for the ground truth and very jagged for the estimation? It appears counter-intuitive, since estimation employs a filter to isolate the noise. It might be interesting here to see what changes by increasing the cut-off frequency (see question above)"
  - Authors: We thank the Referee for this question and provide the following clarification. The difference in smoothness between the ground truth and estimated phase traces originates from both the source signals and the phase shift calculation method. The ground truth phase is derived from LES data by averaging along 36 radial lines and applying a narrow band-pass filter to isolate only the Helix component. We do this for clear comparison, as presence of other frequency components would make comparison rather difficult in the time domain. We then compute the phase using the Hilbert transform of the filtered tilt and yaw wind components. This approach, applied to a clean and spatially averaged harmonic signal, yields a very smooth phase signal.

By contrast, the estimator reconstructs the phase shift from blade load signals using an Extended Kalman Filter. These measurements are affected by turbulence, local unsteady aerodynamics, and structural dynamics, and they are limited to just three points (the blades). The estimator's phase output is inherently noisier and more variable, even though the EKF includes a dynamic model and applies smoothing internally.

Finally, we note that the filtering mentioned by the Referee refers not to a signal filter but to a noise estimation procedure used to construct the covariance matrices in the EKF. The actual input signals to the estimator are not pre-filtered.

Text excerpt: The resulting tilt and yaw wind signals are narrow-band and spatially averaged, leading to clean harmonic components. Applying the Hilbert transform to these filtered signals yields a smooth and consistent phase trace, suitable as a ground truth reference.

The ground truth signal is smoother due to spatial averaging and narrow-band filtering before applying the Hilbert transform. The EKF estimate is reconstructed from noisy blade load data at three locations and therefore exhibits greater variability.

**Algorithm 1 Ground Truth Wind Estimation from LES**

- 1: **Input:** LES velocity field with only T1 present; sampling distance 5D behind T1.
- 2: **Output:** Fixed-frame wind signals  $U_{col}(t)$ ,  $U_{tilt}(t)$ ,  $U_{yaw}(t)$
- 3: Step 1: Define Sampling Geometry
  - 1. Set the number of sampling lines n.
  - 2. Define azimuthal angles  $\psi_{L,1}, \ldots, \psi_{L,n}$  uniformly around the rotor disk center of T2.
  - 3. For each  $\psi_{L,i}$ , define a line extending radially outward from the rotor center at 5D downstream (see Fig. 10).

**4: Step 2: Compute Line-Averaged Velocities**

- 1. For each time step t and each line i = 1 to n:
  - (a) Sample the LES wind velocity  $U_{\text{line},i}(t,s)$  along the spatial coordinate s.
  - (b) Average along s to obtain  $U_{L,i}(t) = \text{mean}_s(U_{\text{line},i}(t,s))$ .

**5: Step 3: Apply Generalized Coleman Transform**

1. For each time step t, compute:

$$U_{\text{col}}(t) \leftarrow \frac{1}{n} \sum_{i=1}^{n} U_{L,i}(t)$$
$$U_{\text{tilt}}(t) \leftarrow \frac{2}{n} \sum_{i=1}^{n} U_{L,i}(t) \cos(\psi_{L,i})$$
$$U_{\text{yaw}}(t) \leftarrow \frac{2}{n} \sum_{i=1}^{n} U_{L,i}(t) \sin(\psi_{L,i})$$

2. Store the resulting signals as the time series ground truth.

**6: Step 4: Isolate Helix Component**

- 1. Apply a band-pass filter to  $U_{\rm tilt}(t)$  and  $U_{\rm yaw}(t)$  (cutoff chosen  $\approx 0.15$  Hz to capture the Helix frequency).
- 2. Retain  $U_{\rm col}(t)$  unfiltered.
- 7: Return  $U_{col}(t)$ ,  $U_{tilt}(t)$ ,  $U_{yaw}(t)$
- Referee: Section 4.2: "Table 3: For the 180° case, the phase errors on Utilt and Uyaw are very high, but the coherence is very high. This is rather counter-intuitive and deserves further analysis."

Date June 11, 2025 Our reference n/a Page/of 11/15

> Authors: We thank the reviewer for this insightful comment. We agree that the combination of high coherence and large phase error appears counter-intuitive at first glance. However, this behavior originates from the fact that the estimator consistently tracks a periodic signal at the correct frequency (i.e., strong coherence), but with a persistent phase bias.

> > For the 180° case, the estimator maintains a strong linear correlation with the ground truth (as seen in the coherence), but due to limitations in the model structure or the ground truth computation, the phase estimate lags or leads the ground truth phase by a constant offset. This persistent bias increases the absolute phase error but does not affect the coherence metric, which measures consistency in frequency alignment rather than absolute phase alignment.

To clarify this point, we have added a brief explanation in the manuscript (Section 4.2).

- Text excerpt: An interesting comparison can be made between coherence and phase error across the different offset cases. In the 180° case, coherence values are among the highest of all channels, while the corresponding phase errors, especially for the tilt and yaw components, are also relatively large. This suggests that the estimator accurately captures the dominant frequency content but demonstrates a systematic phase offset. Such biases may arise from discrepancies between the internal dynBEM model and the true wake dynamics observed in the LES input or limitations of the ground truth computation methodology. In contrast, the 270° case combines high coherence with low phase error and RMSE values, indicating both accurate and stable tracking. The 0° and 90° cases, by comparison, exhibit lower coherence and higher RMSE, suggesting reduced estimator robustness and increased sensitivity to model mismatch in those configurations.
  - Referee: Section 4.4: "With respect to which case the increase/loss of overall power production are evaluated? I guess the comparison is made between the synchronized case and the BL Helix case, although this is not clearly stated. Anyhow, for the "optimal" case, a power gain of 10% on the third turbine is registered, at the cost of a power loss of 5% on the second. This appears to suggest that: i) with only two turbines, the synchronization leads to a power loss; ii) if the synchronization would be applied also on the third turbine, its power loss would decrease as well, leading to a power decrease. I think these points should be discussed in detail, to make the conclusions about power increase much more realistic."
  - Authors: We thank the reviewer for raising these points. Regarding the first comment, we have implemented textual changes to clarify to which the power production cases are evaluated.

Regarding (i): it is indeed correct that in our setup, the upstream turbine (T2) may incur a small power loss when synchronized with T1. This is by design. The principle behind Helix-based wake mixing control and wake steering strategies is that an upstream turbine modifies its wake (via Helix actuation in our case) to benefit a downstream turbine. This typically comes at a small cost to the actuating turbine, but results in an overall power gain at the wind farm level due to the improved inflow to downstream turbines (see Fig. 13, most right plot). In our optimal phase-offset case ( $\varphi_{off} = 150^{\circ}$ ), T2 loses approximately 1-2% power while T3 gains around 10%, yielding a positive combined effect of 5% as a result of our synchronization strategy. These gains are indeed on top of the gains already provided by having Helix on T1 (BL Helix case).

Regarding (ii): We believe there may be a misunderstanding. If T3 were to apply Helix control synchronized with T2, it would now act as an actuating turbine, not a passive beneficiary. This would likely result in a power loss for T3, not a power recovery, as it would be modifying its own wake to benefit a (hypothetical) downstream turbine (e.g., T4). In other words, applying Helix control to T3 would shift its role from beneficiary to contributor in the synchronization chain, with associated power costs. Implementing such a strategy on T3 would therefore only be beneficial if a T4 were present to receive a wind field with enhanced wake recovery.

We have implemented textual changes to ensure no misunderstandings arise in accordance with the points raised by the reviewer.

Text excerpts: Note that the power differences shown are with respect to the BL Helix case (see Table 2). Hence, the power increases are on top of the power increases *already* generated by implementing the Helix on the upstream turbine.

It appears that, from this figure, the optimal offset is at  $150^{\circ}$ , which yields a power gain of around 10% on the third turbine (middle plot). Collectively, this amounts to a power increase of around 5% (right plot), since a small loss of 1-2% (left plot) on T2 can be observed. This illustrates a core principle of wake mixing control: the upstream turbine (here, T2) may incur a small power loss to improve the inflow to a downstream turbine (T3), resulting in a net farm-level gain. Such upstream sacrifices are typical of coordinated wind farm flow control [6].

It is worth noting that extending synchronization to T3 would change its role from passive beneficiary to active contributor. By doing so, T3 would likely experience a small power loss, as it would then act to enhance the inflow to a hypothetical fourth turbine. Hence, further actuation downstream is only beneficial if additional turbines can exploit the modified wake.

Date June 11, 2025 Our reference n/a Page/of 13/15

- Referee: Section 4.5: "There is no mention of the Coriolis effect on the wake, so I am not sure whether this effect is taken into account also in the turbine simulation, or only on the precursor one. It is also not clear whether the inflow velocity profile has a veer at the hub height or not. Showing the time-average and rms of the inflow velocity profiles can be useful here, as well as discussing the effect of the veer on the wake (see for instance Manganelli et al. "The effect of Coriolis force on the coherent structures in the wake of a 5MW wind turbine" 2025), which should in fact deform the helix."
- Authors: We thank the reviewer for the valuable suggestions. The Coriolis force is included in the precursor simulation via geostrophic forcing and influences the inflow to the turbine domain through concurrent precursor coupling. As such, veer and directional shear are implicitly captured in the turbine inflow.
  To clarify this, we have added a time-averaged inflow profile at plane 2 rotor diameters upstream, showing the vertical variation in streamwise velocity, veer, and turbulence intensity. These results are now included in Section 3.1 and shown in the new Fig. 8.
- **Text excerpt:** The vertical profile of the streamwise velocity, along with the lateral veer and turbulence intensity, is shown in Fig. 9. These inflow characteristics result in veered and vertically sheared wakes, contributing to a more realistic simulation environment.

Date June 11, 2025 Our reference n/a Page/of 14/15

**References**

- Nikhar J. Abbas, D Zalkind, Rafael M Mudafort, Gustavo Hylander, Sebastiaan Mulders, David Heff, and Pietro Bortolotti. Nrel/rosco: Raaw v1.2, May 2022. URL https://doi.org/10.5281/zenodo.6543598.
- [2] R J Barthelmie, K. Hansen, S. T. Frandsen, O. Rathmann, J. G. Schepers, W. Schlez, J. Phillips, K. Rados, A. Zervos, E. S. Politis, and P. K. Chaviaropoulos. Modelling and measuring flow and wind turbine wakes in large wind farms offshore. *Wind Energy*, 12(5):431–444, 2009. doi: https://doi.org/10.1002/we. 348. URL https://onlinelibrary.wiley.com/doi/abs/10.1002/we.348.
- Joeri A. Frederik and Jan Willem van Wingerden. On the load impact of dynamic wind farm wake mixing strategies. *Renewable Energy*, 194: 582-595, 2022. ISSN 0960-1481. doi: https://doi.org/10.1016/j.renene. 2022.05.110. URL https://www.sciencedirect.com/science/article/pii/ S0960148122007613.
- [4] Joeri A. Frederik, Bart M. Doekemeijer, Sebastiaan P. Mulders, and Jan Willem van Wingerden. The helix approach: Using dynamic individual pitch control to enhance wake mixing in wind farms. *Wind Energy*, 23(8):1739–1751, 2020. doi: https://doi.org/10.1002/we.2513. URL https://onlinelibrary.wiley.com/ doi/abs/10.1002/we.2513.
- [5] H. Korb, H. Asmuth, and S. Ivanell. The characteristics of helically deflected wind turbine wakes. *Journal of Fluid Mechanics*, 965:A2, 2023. doi: 10.1017/ jfm.2023.390.
- [6] J. Meyers, C. Bottasso, K. Dykes, P. Fleming, P. Gebraad, G. Giebel, T. Göçmen, and J. W. van Wingerden. Wind farm flow control: prospects and challenges. *Wind Energy Science*, 7(6):2271–2306, 2022. doi: 10.5194/ wes-7-2271-2022. URL https://wes.copernicus.org/articles/7/2271/ 2022/.
- [7] Carl Edward Rasmussen and Hannes Nickisch. Gaussian processes for machine learning (gpml) toolbox. *Journal of Machine Learning Research*, 11:3011-3015, 2010. URL http://www.jmlr.org/papers/v11/rasmussen10a.html.
- [8] Richard J. A. M. Stevens, Dennice F. Gayme, and Charles Meneveau. Coupled wake boundary layer model of wind-farms. *Journal of Renewable and Sustainable Energy*, 7(2):023115, 03 2015. ISSN 1941-7012. doi: 10.1063/1.4915287. URL https://doi.org/10.1063/1.4915287.
- [9] A A W van Vondelen, D C Van Der Hoek, S T Navalkar, and J W Van Wingerden. Synchronized dynamic induction control: An experimental investigation. volume 2767, page 032027. IOP Publishing, jun 2024. doi: 10.1088/ 1742-6596/2767/3/032027. URL https://dx.doi.org/10.1088/1742-6596/ 2767/3/032027.

Page/of 14/15

Date June 11, 2025 Our reference n/a Page/of 15/15

> [10] Aemilius A. W. van Vondelen, Sachin T. Navalkar, Daan R. H. Kerssemakers, and Jan Willem Van Wingerden. Enhanced wake mixing in wind farms using the helix approach: A loads sensitivity study. In 2023 American Control Conference (ACC), pages 831–836, 2023. doi: 10.23919/ACC55779.2023.10155965.